## Registered report

psychology/human-computer interaction

loot boxes, video games, video gaming regulation, information
technology and interactive entertainment law, consumer protection, industry self-regulation and social corporate responsibility

**Author for correspondence:**
Leon Y. Xiao
e-mails: lexi@itu.dk, leon.xiao@york.ac.uk

## PUBLISHING

# Beneath the label: unsatisfactory compliance with ESRB, PEGI and IARC industry self-regulation requiring loot box presence warning labels by video game companies

Leon Y. Xiao[1,2,3,4,5]

[1]Center for Digital Play, IT University of Copenhagen, København, Denmark
[2]Department of Computer Science, University of York, York, UK
[3]School of Law, Queen Mary University of London, London, UK
[4]Transatlantic Technology Law Forum, Stanford Law School, Stanford University, Stanford, CA, USA
[5]The Honourable Society of Lincoln's Inn, London, UK

LYX, 0000-0003-0709-0777

Loot boxes in video games are a form of in-game transactions with randomized elements. Concerns have been raised about loot boxes' similarities with gambling and their potential harms (e.g. overspending). Recognizing players' and parents' concerns, in mid-2020, the Entertainment Software Rating Board (ESRB) and PEGI (Pan-European Game Information) announced that games containing loot boxes or any forms of in-game transactions with randomized elements will be marked by a new label stating 'In-Game Purchases (Includes Random Items)'. The same label has also been adopted by the International Age Rating Coalition (IARC) and thereby assigned to games available on digital storefronts, e.g. the Google Play Store. The label is intended to provide more information to consumers and allow them to make more informed purchasing decisions. This measure is not legally binding and has been adopted as industry self-regulation or corporate social responsibility. Previous research has suggested that industry self-regulation might not be effectively complied with due to conflicting commercial interests. Compliance with the ESRB's, PEGI's and IARC's loot box presence warning label was assessed in two studies. The first study found that 60.6% of all games labelled by

either the ESRB or PEGI (or 16.1% using a more equitable methodology) were not labelled by the other. The majority of the inconsistencies were caused by the ESRB refusing to apply the measure retroactively. Five instances where one age rating organization culpably failed to accurately identify loot box presence were identified (although only two cases were admitted by the relevant organization). Generally, with newly released games, consumers can rely on the PEGI and ESRB labels. PEGI has retroactively labelled many older games, meaning that consumers can expect the labelling to be accurate. However, due to the ESRB's policies (which it has refused to improve), North American consumers cannot rely on the label for many older games containing loot boxes, unlike their European counterparts. The data suggest that the loot box issue is far more pressing on mobile platforms than console/PC platforms. The second study found that 71.0% of popular games containing loot boxes on the Google Play Store (whose age rating system is regulated through IARC) did not display the label and were therefore non-compliant. The IARC's current policy on the Google Play Store is that only games submitted for rating after February 2022 are required to be labelled. This policy (which the IARC has refused to improve) means that most popular and high-grossing games can be, and presently are, marketed without the label, thus significantly reducing the measure's scope and potential benefit. The Apple App Store still does not allow loot box presence to be disclosed. At present, consumers and parents cannot rely on this self-regulatory measure to provide accurate information as to loot box presence for mobile games. Due to their immense scale, the mobile markets pose regulatory and enforcement challenges that PEGI admits are not yet resolved. The mere existence of this measure cannot be used to justify the non-regulation of loot boxes by governments, given the poor compliance and doubtful efficacy (even if when complied with satisfactorily). Improvements to the existing age rating systems are proposed. Preregistered Stage 1 protocol: https://doi.org/10.17605/OSF.IO/E6QBM (date of in-principle acceptance: 12 January 2023).

# 1. Introduction

Paid loot boxes are products within video games that players buy to obtain randomized rewards [1,2]. Some loot boxes are 'non-paid' and can be obtained without spending real-world money; however, the present study focuses on *paid* loot boxes. Hereinafter, 'loot boxes' refers to *all* forms of randomized video game monetization methods, i.e. any 'in-game transactions with randomized elements' [3]. Concerns have been raised about loot boxes' similarities with gambling and the risks that consumers might overspend money and experience harm [4–9]. Children and other vulnerable consumers (e.g. people experiencing problem gambling issues) might be at particular risk of harm [10,11]. Many countries are considering imposing legal regulation, and a few countries have already taken regulatory actions [12–16]. However, in most countries at present, paid loot boxes are specifically regulated only through industry self-regulation [17]. There are two prominent loot box self-regulatory measures: probability disclosures and text-based warning labels attached to age ratings.

The Apple App Store, similar to many other hardware and software platforms [18], imposes the self-regulatory requirement that all games available on that platform 'offering "loot boxes" or other mechanisms that provide randomized virtual items for purchase' [19] must disclose the probabilities of obtaining those items to customers prior to purchase. Xiao *et al.* assessed companies' compliance with Apple's self-regulatory measure among the 100 highest-grossing iPhone games in the UK and found that only 64% of games containing loot boxes disclosed probabilities. This compliance rate was significantly lower than the 95.6% observed in Mainland China where probability disclosures were (and continue to be) required by law [20].

The second self-regulatory measure is to prewarn players about the presence of loot boxes. The Entertainment Software Rating Board (ESRB), established by the Entertainment Software Association (ESA), reviews the content of video games and provides age ratings depending on the inclusion of certain material, e.g. the amount and degree of violence and sexual content [21]. The ESRB is adopted in North America. PEGI (Pan-European Game Information) performs a similar function in Europe generally [22]. Recognizing the concerns that have been raised about loot boxes, on 13 April 2020, the ESRB and PEGI announced that they will attach an additional text-based warning to the age ratings of video games containing loot boxes [23]. The ESRB uses the 'In-Game Purchases (Includes Random Items)' 'interactive element' [3] (figure 1), while PEGI originally proposed to use the 'In-game Purchases (Includes Paid Random Items)' 'content descriptor' (figure 2) [24]. PEGI secretly changed

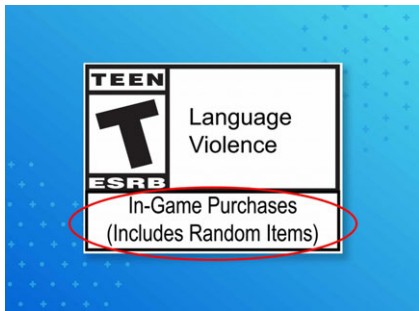

**Figure 1.** The ESRB 'In-Game Purchases (Includes Random Items)' interactive element. © 2020 Entertainment Software Rating Board (ESRB).

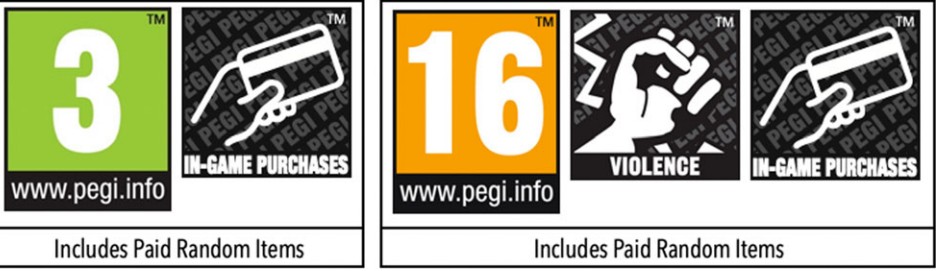

**Figure 2.** The originally announced, but since replaced, PEGI 'In-game Purchases (Includes Paid Random Items)' content descriptor. © 2020 Pan-European Game Information (PEGI).

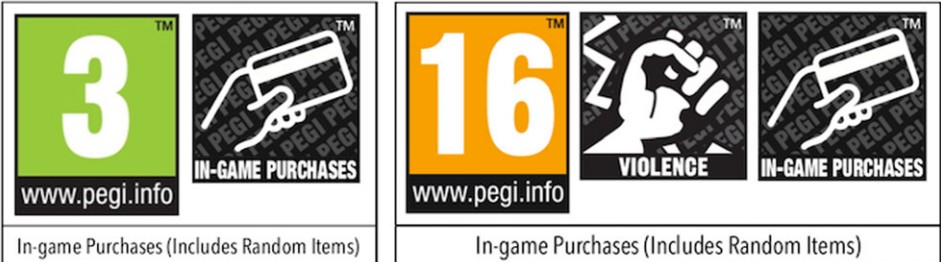

**Figure 3.** The current PEGI 'In-game Purchases (Includes Random Items)' content descriptor. © 2020 (Pan-European Game Information (PEGI).

its label to instead read 'In-game Purchases (Includes Random Items)' (figure 3), which is identical to the ESRB's (except for the capitalization of the 'G'), soon after the initial announcement (without a further announcement) by retroactively partially changing the initial announcement (see §4.3.3 for further detail). As of 16 January 2023, the PEGI announcement's text still referred to the older label, but the image accompanying has been changed to reflect the current (i.e. the ESRB's) label. These two largely identical labels are intended to cover, according to the ESRB, 'all transactions with randomized elements' [3]. The ESRB and PEGI both consciously chose to specifically *not* use the term 'loot boxes' to 'avoid confusing consumers' [3], particularly parents who might not have sufficient knowledge about video games or 'ludoliteracy'.

According to the ESRB, its label accounts for:

'… loot boxes and all similar mechanics that offer random items in exchange for real-world currency or in-game currency that can be purchased with real money' [3].

According to PEGI, its label covers:

'… all in-game offers to purchase digital goods or premiums where players don't know exactly what they are getting prior to the purchase (e.g. loot boxes, card packs, prize wheels)' [24].

These definitions accord with the wide definition for 'loot boxes' adopted by the present study. These labels were intended to 'provide the additional information if the game features paid random items' [24],

such that '…consumers can make more informed decisions when purchasing or downloading a game, instead of finding out after the fact' [3]. Notably, the presence of these labels, or rather the presence of loot boxes, does not affect a game's age rating, because neither the ESRB nor PEGI recognizes loot boxes as actual 'gambling' or 'simulated gambling' [25,26]. These labels can therefore attach to games containing loot boxes but are rated suitable for young children (i.e. ESRB's 'E' or 'Everyone' rating and PEGI's 'PEGI 3' rating) [24]. This is unlike how other content, such as depiction of 'realistic violence', 'illegal drugs, alcohol or tobacco' or 'simulated gambling', would (in certain situations, automatically [27]) cause the game to attract higher age ratings [28,29]. I have previously criticized the labels for not providing sufficient information to truly help players and parents make more informed purchase decisions [23]. The labels fail to identify and explain where and how the loot boxes in a specific game can be purchased, and so players and children cannot easily actively avoid engaging with the mechanics. The labels also do *not* signify whether or not the relevant mechanic provides rewards that can then be transferred to other players and 'cashed-out' [30] (i.e. have real-world monetary value), which is a relevant consideration for many gambling regulators [9,16,17]. The labels might be of some assistance by providing information at the initial point of purchasing or downloading the game; however, once the player has begun playing the game, the labels are no longer helpful. An improvement might be to specifically describe the loot box mechanics to help players actively avoid them and to provide a choice in the options menu to turn the ability to purchase loot boxes on or off (potentially even with the default option set to 'off'). Through experimental studies, Garrett *et al.* have concluded that these labels fail to adequately warn consumers about the potential risks involved with loot boxes and therefore 'fail to adequately inform consumer spending decisions' [31].

The ESRB's and PEGI's wide definitions for 'in-game transactions with randomized elements' [3] and what the present study refers to as 'loot boxes' are effectively identical, despite trivial variations in the wording of the definitions and of the labels. Therefore, the reasonable expectation is that a game containing loot boxes should be labelled with the 'In-Game Purchases (Includes Random Items)' interactive element after being rated by the ESRB in North America and with the 'In-game Purchases (Includes Paid Random Items)' content descriptor or its newer variation after being rated by PEGI in Europe. The ESRB and PEGI should be *consistent* when deciding whether a game contains loot boxes. If one of them fails to label a game with the loot box warning when the other has done so, then the former has highly likely inaccurately rated said game's loot box presence by failing to identify it. The only highly unlikely exception being that a game potentially has separate North American and European versions and only one of which contained loot boxes: such a situation has never been popularly reported.

Research Question 1: Are video games being consistently given the loot box self-regulatory warning label by the ESRB and PEGI?

Hypothesis 1: All games that have been labelled with the 'In-Game Purchases (Includes Random Items)' interactive element by the ESRB should also have been labelled with the 'In-game Purchases (Includes Paid Random Items)' content descriptor or its newer variation by PEGI and vice versa.

The ESRB and PEGI only play a direct role when rating physically published games and are only *indirectly* involved in the rating of each individual digitally released game. Both the ESRB and PEGI are 'participating rating authorities' of the IARC (International Age Rating Coalition), which is a simplified system that allows game companies to simultaneously obtain multiple age ratings for use in different territories for digitally delivered games [32]. After companies fill in a single questionnaire about their games' content, the IARC will produce age ratings that 'also include content descriptors and interactive elements, identifying games and apps that [*inter alia*] offer in-app/game purchases (*as well as those that are randomized*)' (emphasis added) [32]. Specifically, the IARC uses the 'In-Game Purchases (Includes Random Items)' interactive element, which is the ESRB's label and whose wording differs slightly from that of the PEGI label. The IARC is not implemented on the Apple App Store (which uses its own age rating system [33]) but is adopted by the Google Play Store and other major platforms [32]. Depending on which national version of the Google Play Store is visited, the appropriate age rating for that territory is shown. For example, for the game *Guns of Glory* (FunPlus, 2017), the US Google Play Store displays the ESRB rating of Everyone 10+ (https://play.google.com/store/apps/details?id=com.diandian.gog&hl= en&gl=us), while the Danish Google Play Store displays the PEGI rating of 7 (https://play.google.com/ store/apps/details?id=com.diandian.gog&hl=en&gl=dk). *Guns of Glory* has previously been identified as containing loot boxes in multiple studies [18,34,35]. Indeed, the IARC has attached the 'In-Game Purchases (Includes Random Items)' label to the game on both the US and Danish Google Play Stores alongside the respective ESRB and PEGI age ratings.

Notably, the IARC explains that 'Interactive Elements are assigned *universally*, providing notice about the ability to make in-game purchases (including randomized ones)…' (emphasis added) [36]. This contrasts with the IARC's assignment of 'Age Rating and Content Descriptors', which will differ by region [36]. In other words, a game containing loot boxes can receive different age ratings in different regions under the IARC system, but the loot box warning label, which is an interactive element, should be attached to that game regardless of region. Universal or global assignment of the label means that the IARC has effectively extended the 'jurisdictional' scope of the loot box self-regulatory warning label requirement to countries beyond those covered by the ESRB and PEGI. For example, Germany, despite being in Europe, does not use PEGI and instead adopts the alternative USK (Unterhaltungssoftware Selbstkontrolle (USK) to provide age ratings. Up until 31 December 2022 (inclusive), the USK did not assess the presence of loot boxes and did not require the use of a label to signify their presence in relation to physical games marketed in Germany [37]. (On 14 December 2022, the USK announced that it will begin to assess the presence of loot boxes and attach the 'In-Game-Käufe + zufällige Objekte [In-game purchases + random items]' label accordingly to new game submissions from 1 January 2023 [38]). Notably, as of 16 December 2022, the official USK web page explaining the integration of the USK age ratings within the IARC system has *not* been updated to explain that the IARC will now additionally attach 'In-Game-Käufe (zufällige Objekte möglich) [In-Game Purchases (Includes Random Items)]' to games containing loot boxes and, instead, the web page still only states that games allowing for additional in-game purchases will be attached with the generic 'In-Game-Einkäufe [In-Game Purchases]' [39]. However, the USK is a 'participating rating authority' of the IARC [32], and so, even before the USK introduced its own loot box warning label, games containing loot boxes were being attached with 'In-Game-Käufe (zufällige Objekte möglich)' on the German Google Play Store: for example, *Guns of Glory* (https://play.google.com/store/apps/details?id=com.diandian.gog&hl=de&gl=de) as of 18 September 2022.

Draft laws in the US that have failed to pass [40] tried to require games containing loot boxes to 'prominently disclose to the consumer at the time of … purchase a bright red label that is easily legible and which reads: "Warning: contains in-game purchases and gambling-like mechanisms which may be harmful or addictive"' [41,42]. Another (since died) Bill intended to require the following (arguably not entirely scientifically inaccurate) loot box warning label within the US state of Illinois:

> 'Attention Parents: A Loot Box System exists in this game that permits an unlimited amount of REAL MONEY to be spent without any age restriction. REAL MONEY is exchanged for random digital items. This process has been linked to REAL LIFE GAMBLING ADDICTIONS in both children and adults. Please regulate your own spending as well as your children's spending' [43].

Other countries might be considering imposing similar information-based warning labels to address the potential harms of loot boxes. Previous research has found that other industries, such as alcohol [44], tobacco [45] and gambling [46], have all taken various corporate actions that probably reduced the effectiveness of product warnings. Loot box probability disclosures are known to have been implemented suboptimally by video game companies: specifically, lacking prominence and being difficult to access [18,20]. Compliance with Belgium's 'ban' on loot boxes through applying pre-existing gambling law has also been poor [35].

When filling in the content rating questionnaire, Google warns that: 'Misrepresentation of your app's content may result in removal or suspension, so it is important to provide accurate responses to the content rating questionnaire'. The IARC also recognizes that ratings can be changed through 'post-release modification' [47] and states that: 'IARC rating authorities [*inter alia*, the ESRB and PEGI] monitor ratings assigned to games and apps to ensure accuracy. Corrections, if needed, are implemented promptly by storefronts'. However, considering prior research, reasonable doubt must be cast on the compliance rate with the self-regulatory requirement of attaching loot box warning labels.

Rather than to assess the 100 presently highest-grossing Google Play Store games as to whether they contain loot boxes (as previous studies have done [18,20,34,35]) and then to check whether they are displaying the label, it is more economical and efficient to instead examine games previously known to contain loot boxes. If a game that was known to contain loot boxes is displaying the label, then it is no longer necessary to assess whether said game still contains loot boxes through gameplay, as this can be reasonably assumed. Only those games previously known to contain loot boxes but are not displaying the label need to be reassessed through gameplay. This expediency is desirable because it is hoped that the present study's results could be published promptly and thereby contribute to the efforts of the UK Government's Department for Digital, Culture, Media & Sport's technical working group that is developing industry self-regulation for loot boxes with the aim of reducing harm [48].

The sample selection (as detailed below) was based on previously highest-grossing games (many of which probably still remained high grossing and popular presently) [18,20,34,35]. This, therefore, represents a sample of particular interest for players, parents, policymakers and the age-rating organizations. However, some limitations should be noted. Firstly, the compliance rate among this sample of historically (and potentially presently) high-grossing games is not necessarily representative of that of financially worse performing games (which might be less scrutinized by players and other companies and therefore less likely to comply or, contrastingly, might be performing worse financially because they have accurately displayed the label) or the overall situation on the Google Play Store. Secondly, these games were previously highlighted in published academic work as having contained loot boxes [18,20,34,35], and, therefore, their operating companies might have since become more likely to comply (when compared with a newly published game that has not yet gained any notoriety), as companies have reportedly complied with the Belgian 'ban' on loot boxes only following the publication of Xiao and media reporting thereof [35,49] and 4 years after they were originally supposed to have done so.

Research Question 2: Are video games previously known to be high grossing and contain loot boxes and presently containing loot boxes on the Google Play Store accurately displaying the IARC 'In-Game Purchases (Includes Random Items)' label?

Hypothesis 2: All titles in the present sample of video games previously known to contain loot boxes, and which are presently available on the Google Play Store and continue to contain loot boxes, will accurately display the IARC 'In-Game Purchases (Includes Random Items)' label.

The present series of two studies did not seek to assess the efficacy of the loot box self-regulatory labels on consumer behaviour (see [31]) and instead sought to assess (i) whether the ESRB and PEGI have *consistently* applied the loot box self-regulatory warning label and (ii) whether companies have complied with this self-regulation by *accurately* labelling games containing loot boxes with the relevant notice.

# 2. Method

## 2.1. Study 1

The ESRB provides a public search tool for identifying the age ratings, content descriptors and, importantly for Study 1, interactive elements, including the 'In-Game Purchases (Includes Random Items)' label, for specific games [50]. However, it is not possible to use the search tool to specifically identify only games with the 'In-Game Purchases (Includes Random Items)' label. Using the relevant filter for the label unhelpfully brings up all games with 'No Interactive Elements' (the overwhelming majority) and those with the relevant label. The ESRB also publishes a list of all games that it has rated in reverse chronological order [51]. By using the 'Refine Search' function of the search tool and limiting the 'Time Frame' to 'Past Year' (the longest period that could be chosen) and applying no other filters, a list of all games that were rated in the year leading up to 21 September 2022 were extracted through data scraping. This list consisted of 698 individual entries (a few games appeared as multiple entries because different editions and platforms were sometimes rated and listed separately). In total, 21 entries (3.0%) were labelled by the ESRB with the 'In-Game Purchases (Includes Random Items)' interactive element. Two entries were excluded for bearing the exact same name as another entry. A third entry was excluded because although it bears an additional subtitle (*FIFA 22 Legacy Edition*), it is the same game as another entry (*FIFA 22*) and appears to have probably been rated on the same date. A list of 18 individual video game titles that were labelled by the ESRB with the loot box self-regulatory warning in the year leading up to 21 September 2022 was thereby produced. Based on how many games appeared as results when the Time Frame filter was set to 'Past Year', it can be estimated (appreciating that seasonable variability and COVID-19 impacts cannot be accounted for) that the ESRB rated approximately 700 games per year historically. This information can be used to infer that the ESRB rated approximately 992 games in the 17 months between 13 April 2020 (the date on which the labels were announced and began to be assigned) and 21 September 2021 (the date after which the list of games rated in the past year leading up to 21 September 2022 started). The 1415 entries (amending and improving on the preregistered 1000 entries) that immediately preceded the 698 entries that have already been collected on the reverse chronological order list were also collated through data scraping. An additional 26 entries labelled by the ESRB with the 'In-Game Purchases (Includes Random Items)' interactive element were identified, and 10 entries bearing the same or a

substantially similar name were excluded as above. These additional 16 entries were combined with the 18 previously identified entries to form an approximately complete list of 34 games that have been labelled by the ESRB with the loot box self-regulatory warning since 13 April 2020 (hereinafter, the 'ESRB List'). The ESRB List was generated thusly because it was deemed impractical to analyse all 31 636 individual historical entries (existing on 21 September 2022) and the ESRB provided no information as to the exact date that a rating was given, besides allowing an inference to be drawn through the Time Frame filter. Certain games are also published months after a rating has been granted, so the release date of games also cannot be used to determine the relevant rating date. It was deemed unwise and potentially leading to a conflict of interest (and a change in compliance behaviour) to contact the ESRB and ask for a complete list of games that it has labelled with the warning, although this might be done following the publication of the present study.

PEGI similarly provides a search tool for identifying the age ratings and content descriptors (including the 'In-game Purchases (Includes Random Items)' label) for specific games [52]. Unlike the ESRB search tool, the PEGI search tool *can* be used to produce a list of all games ever rated by PEGI that were given the 'In-game Purchases' content descriptor, if the 'DESCRIPTOR' of 'In-Game Purchases' is selected in the 'EXTENDED SEARCH' options [53]. The 'In-game Purchases (Includes Random Items)' is treated as a subtype of the overarching 'In-game Purchases' content descriptor, and therefore all games that have been given the loot box self-regulatory warning are included in said list. On 21 September 2022, a list of 523 individual results of games that have ever been labelled by PEGI with the 'In-game Purchases' content descriptor was produced. Again, a number of games appeared as multiple entries, as different editions and platforms were sometimes rated and listed separately. In total, 125 results (23.9%) were ever labelled by PEGI with the 'In-game Purchases (Includes Random Items)' content descriptor or its older variation. Entries were excluded for bearing the same name as another entry (55 entries) and being the same game as another entry despite minor changes to the title (e.g. '*World of Tanks on PlayStation 4*' as compared with '*World of Tanks*', six entries). A list of all 64 individual video game titles that have ever been labelled by PEGI with the loot box self-regulatory warning was thereby produced (hereinafter, the 'PEGI List').

The following variable was measured:

### 2.1.1. Presence of the loot box self-regulatory warning label on the other system

The games on the ESRB List were entered into the PEGI search tool [52] and vice versa with the PEGI List and the ESRB search tool [50]. Screenshots were taken of the relevant ratings, content descriptors, and/or interactive elements. If the corresponding loot box self-regulatory label could be found for the game on the other age rating system, then this game was marked as 'consistent', but if not, then 'inconsistent'. If a game appeared on both the ESRB List and the PEGI List, then it was deemed 'consistent', but screenshots were taken on both systems to corroborate this. Some reasonable flexibility was allowed when searching for a corresponding game if a game with the exact same title could not be found. Any deviation was recorded. If a game could not be found on the other system even after allowing for a reasonable amount of flexibility with the search term, then it would have been excluded from analysis.

The 'consistency rate' between the ESRB's and PEGI's usage of the loot box warning self-regulation was calculated as follows:

$$\frac{\text{games that have been labelled with the loot box warning by both the ESRB and PEGI}}{\text{(all games on the ESRB and PEGI Lists} - \text{any duplicate or excluded games)}}.$$

Hypothesis 1 would have been accepted had the consistency rate been greater than or equal to 95%. Otherwise, Hypothesis 1 would have been rejected. In terms of the interpretation of results, a consistency rate of greater than or equal to 95% would have been viewed as the ESRB and PEGI having been sufficiently consistent. A consistency rate of greater than or equal to 80% but less than 95% would have been deemed as the self-regulatory measure not having been applied sufficiently consistently by the ESRB and PEGI, and thus the rating processes require improvements to enhance cohesion. A consistency rate of less than 80% would have been seen as the measure having been applied inconsistently, and thus the rating processes being in need of significant improvements. These cut-offs and corresponding potential interpretations were based on the author's own opinion on what is a 'satisfactory' self-regulatory measure and what he deemed most policymakers would agree with.

Study 1 achieved level 3 of bias control as recognized by *Peer Community In Registered Reports* (*PCI RR*), as it was necessary to attempt to collate the ESRB and PEGI Lists to affirm the study's practical

feasibility. I certified in the registered protocol that: at the time, I have 'not yet observed ANY part of the data/evidence' [54], specifically, I have not searched for games on either List using the other rating system's search tool.

## 2.2. Study 2

The sample of 100 (or potentially fewer) games were selected using the following steps:

1. The sample was derived from the samples of four previous studies assessing loot box prevalence among mobile games in different countries, which examined 531 separate instances of video games and identified whether they contained loot boxes [18,20,34,35].
2. Among those 531 games, 100 were originally studied in Chinese and not in English [20]. Those 100 Chinese games were reviewed in 2021 to identify a subset of 31 games that were also then available in English, which were reassessed in a UK study [18]. The present study is less interested with the compliance situation of games available only in Chinese and more concerned with the compliance situation in North America and Europe (i.e. 'Western' countries) where the ESRB and PEGI self-regulate; therefore, those 100 Chinese games were not reviewed again as the previously distilled list of 31 games that were available in both languages were taken into account.
3. A list of 431 games combining the results of three previous studies was collated [18,20,34]. Any duplicates and any games assessed to have *not* previously contained loot boxes were removed. Some reasonable flexibility as to the game's title was allowed when searching for and removing duplicate games (e.g. changes to the subtitle to reflect a content update). Any deviation was recorded. The remaining games therefore formed a list of non-duplicate games that were known to contain loot boxes.
4. It was known that two so-called 'sand box' games (specifically, *Roblox* (Roblox Corporation, 2006) and *Minecraft* (Mojang Studios, 2011)) would be included on that list. These two games contain a significant amount of third-party user-generated content, including loot boxes [18,35]. This represents a particular compliance difficulty as these 'platform' games' developers and publishers would need to ensure not only compliance by themselves but also compliance by many third parties creating content for these games [49,55]. To ensure that both of these games would be assessed, they were removed from the list and did not form part of the sample. Their compliance situation was separately reported. If either game would have become unavailable for download and incapable of being assessed, then this would have been noted in lieu.
5. Therefore, the present study's sample was a total of 100 random games from the list of non-duplicate games that were known to contain loot boxes.
6. Alternatively, had that list contained fewer than 100 games, the entire list would have formed the sample.
7. Had any game in the sample been no longer available for download from the Google Play Store by the data collection period then it would have been excluded from the sample and replaced with another random game from the list. Had that list contained fewer than 100 games or had no games been left on that list to replace the excluded game, then the study would have proceeded with the available games even if the sample was formed of fewer than 100 games.
8. The same exclusion and replacement (if possible) procedure would have applied had *Guns of Glory* been included. This game is specifically being excluded as it has been used as an example to test and illustrate the present study's methodology for the stage 1 registered report submission and its 'results' have already been observed.

The following variables were measured:

### 2.2.1. Presence of the interactive element of 'In-Game Purchases (Includes Random Items)'

The Google Play Store page of the relevant game was reviewed to check whether the IARC interactive element of 'In-Game Purchases (Includes Random Items)' has been noted alongside the game's age rating. The US and Danish Google Play Stores for each game were checked to see whether the label has been attached to both the ESRB and the PEGI ratings, respectively. A simple change of the parameter 'gl = [country code]' in the game's Google Play Store URL allowed for the switching of regions. The country code for the US is 'us', while Denmark uses 'dk'. To illustrate using the example of *Guns of Glory,* the US store could be visited through the following URL: https://play.

google.com/store/apps/details?id=com.diandian.gog&hl=en&gl=us, while the Danish store could be visited through: https://play.google.com/store/apps/details?id=com.diandian.gog&hl=en&gl=dk. PDF printouts of the relevant web pages (showing the URL visited) were made.

### 2.2.2. Presence of paid loot boxes (newly assessed)

If the Google Play Store page of a game known to previously contain paid loot boxes did not show the IARC interactive element of 'In-Game Purchases (Includes Random Items)' alongside the game's age rating, then that game was played for up to an hour to identify whether paid loot boxes were still being implemented and sold in that game. Any identified paid loot boxes had a screenshot taken. If a paid loot box could not be identified within that timeframe, then the game would have been coded as not containing paid loot boxes.

To align with the methodology of prior studies [18,20,35], a 'paid loot box' was defined as being either an Embedded-Isolated random reward mechanism (which are video game mechanics that players must pay real-world money to activate and which provide randomized rewards that do *not* possess direct real-world monetary value) or an Embedded-Embedded random reward mechanism (whose activation also must be paid for by players with real-world money but which *do* provide randomized rewards that possess direct real-world monetary value), as defined by Nielsen & Grabarczyk [4].

In particular, it is emphasized that so-called 'social casino games' or 'simulated casino games', in which the player is able to spend real-world money to participate in simulated traditional gambling activities (i.e. 'games of chance' or 'mixed games of chance and skill'; e.g. slot machines, poker and blackjack) and win or lose virtual currency *randomly* [35], were counted as games containing 'loot boxes' for the purposes of Hypothesis 2, despite some debate on that point within the academic literature [56,57]. This is because spending real-world money to participate in a social casino game constitutes an in-game '[transaction] with randomized elements', per the ESRB's definition [3]. The present study's definition of 'paid loot box' encompasses both mechanics that are commonly known as 'loot boxes' and social casino games. This accords with both the ESRB's and PEGI's definitions for mechanics that the loot box warning labels are supposed to cover [3,24]. However, the relevant compliance rates (see below) among 'social casino games' (which were identified using the definition above) and non-'social casino games' were additionally separately reported to provide nuance.

Further, again aligning with the methodology of prior studies [18,35], so-called 'sand box' games, such as *Minecraft* or *Roblox*, that contain a significant amount of third-party user-generated content were assumed to contain paid loot boxes without the need for such a mechanic to be specifically identified and a screenshot taken.

### 2.2.3. Date and time of data collection

The date and time, based on Greenwich Mean Time, on and at which the interactive element and paid loot boxes was searched for, was recorded.

Inter-rater reliability through dual-coding was not calculated because the methodology has been repeatedly used and refined and is known to be reliable [58]. The raw data and a full library of PDF printouts and screenshots showing, *inter alia*, the relevant Google Play Store web page sections and in-game loot box purchase pages for each game has been made available via https://doi.org/10.17605/OSF.IO/YZKUP for public scrutiny.

The 'compliance rate' with the loot box warning self-regulation was calculated as follows:

$$1 - \frac{\text{games newly assessed as containing loot boxes but not displaying the interactive element}}{\begin{array}{c}\text{(all games previously known to previously contain loot boxes} \\ - \text{ games newly assessed as not containing loot boxes)}\end{array}}.$$

Hypothesis 2 would have been accepted had the compliance rate been greater than or equal to 95%. Otherwise, Hypothesis 2 would have been rejected. In terms of the interpretation of results, a compliance rate of greater than or equal to 95% would have been viewed as the self-regulatory measure having been nearly perfectly complied with and worthy of commendation. A compliance rate of greater than or equal to 80% but less than 95% would have been deemed as the self-regulatory measure having been mostly complied with, although improvements are needed. A compliance rate of less than 80% would have been seen as the measure having *not* been adequately complied with and in need of significant improvements to achieve its regulatory aim. Again, these cut-offs and corresponding potential

interpretations were based on the author's own opinion on what is a 'satisfactory' self-regulatory measure and what he deemed most policymakers would agree with.

Study 2 achieved level 6 of bias control as recognized by *PCI RR* as the relevant data did not yet exist at the relevant time [54].

The sample sizes for both studies were justified on the basis of resource constraints: specifically, the researcher had limited time and was seeking to promptly complete the study in time to assist in the government-supported, industry self-regulatory efforts regarding loot boxes contemporaneously under way in the UK [48].

In accordance with the *Danish Code of Conduct for Research Integrity* [59], as adopted by the IT University of Copenhagen, the present series of two studies did not require research ethics assessment and approval because no human participants or personal data were involved and only publicly available information was examined and recorded.

# 3. Results

## 3.1. Study 1: ESRB and PEGI consistency

### 3.1.1. Confirmatory analysis

The ESRB and PEGI Lists combined to form a list of 98 separate entries. Among those, 24 entries were combined into another entry because they appeared on both the ESRB and PEGI Lists; referred to the same underlying game; and so represented cases where both age rating systems attached the label. Of the remaining 74 different games that were attached with the loot box warning label by either the ESRB or PEGI up until 28 September 2022, 24 games (32.4%) were labelled thusly by both, 10 games (13.5%) were labelled only by the ESRB and 40 games (54.1%) were labelled only by PEGI, as shown in table 1.

A closer examination was made of the 50 games regarding which the ESRB and PEGI did not conform and apparently came to different conclusions as to whether or not said game contained loot boxes and therefore should be attached with the label on 21 September 2022. PEGI rated two of those games (4.0%) after 21 September 2022 and correctly labelled them, so these are treated as cases of consistency. A total of eight games (16.0%) were submitted to only one of the two bodies: five games (10.0%) were submitted only to PEGI and three games (6.0%) were submitted only to the ESRB. Given that the other age rating organization did not have an opportunity to examine, and could never have potentially labelled, these eight games, they were excluded from the consistency rate calculation as preregistered.

In summary, among 66 games that both the ESRB and PEGI rated and therefore had an opportunity to attach the loot box presence warning label to, the ESRB and PEGI were genuinely inconsistent in relation to 40 games (60.6%) and consistent with 26 games, which means that the 'consistency rate' was calculated to be only 39.4%. Among the 40 disagreements, 35 (87.5%) were attributed to the ESRB failing to attach the label while five (12.5%) were attributed to PEGI.

Hypothesis 1 is rejected as the consistency rate of 39.4% is less than 95%.

### 3.1.2. Exploratory analysis

It became apparent during the data collection process that PEGI applied the label to some games that were originally rated prior to the announcement of the label *retroactively*, while the ESRB did not. This is demonstrated by the labelling status of the *FIFA* games (Electronic Arts, 1993–present), which were released through annual editions: PEGI has labelled *FIFA 15* to *FIFA 22* with the label. By contrast, the ESRB failed to label *FIFA 15* to *FIFA 20*, but did label *FIFA 21* and *FIFA 22*. The *FIFA* games are always released in the year before that edition's numbering, meaning that *FIFA 20* was released on 27 September 2019 and *FIFA 21* was released on 9 October 2020 (according to PEGI, as shown in the relevant PEGI Search Tool Printouts at the data deposit link[1]). The label was announced by the ESRB and PEGI on 13 April 2020. It would be reasonable to conclude from these dates that the ESRB applied the label only after the label announcement date, but PEGI has gone back and rerated at least some games to attach the label (as both organizations have since confirmed). As further detailed in §4.1.1 of the Discussion section, either approach was justifiable given the wording of the

---

[1]Hereinafter, any references to a game's release date according to PEGI can also be corroborated by relevant printouts at the data deposit link.

**Table 1.** Synthesis and interpretation of the official explanations for the disagreements/inconsistencies ($n = 40$). Note: underlined text means the author is of the view that the relevant age rating organization is culpable for some failings. The wording of 'et al.' refers to when a title was submitted as multiple entries and some of those entries may have additional subtitles attached to the original game title.

| game title and/or category description | label missing from... | ESRB labelling status or response | PEGI labelling status or response |
|---|---|---|---|
| 29 games originally rated prior to the label being introduced | ESRB | not responsible for updating the label | duly labelled |
| | | did not invite publishers to add the label | invited publishers to add the label |
| | | the publisher cannot add labelling in the search tool even if desired | relevant publisher voluntarily added the label upon invitation |
| | | did 'encourage' publishers to 'add [label] to packaging and marketing materials if applicable' | |
| *Mobile Suit Gundam Battle Operation 2* | PEGI | duly labelled | rated prior to the label being introduced |
| | | | invited publishers to add the label |
| | | | relevant publisher did *not* voluntarily add the label upon invitation |
| | | | the label has not yet been added as of 12 February 2023. |
| *UFC 4* | ESRB | does not contain loot boxes | agrees does not contain loot boxes |
| | | | publisher disclosed out of an abundance of caution |
| *PGA Tour 2K21* | | | technically a false positive |
| | | | since corrected by removing the label |

(Continued.)

**Table 1.** (*Continued.*)

| game title and/or category description | label missing from... | ESRB labelling status or response | PEGI labelling status or response |
|---|---|---|---|
| *Arcadegeddon* | PEGI | labelled based on self-reporting possibly technically a false positive | does not contain loot boxes |
| *Riders Republic* | | | |
| *Blankos Block Party* | PEGI | labelled based on self-reporting | does not contain loot boxes in PEGI territories due to region-specific geo-blocking |
| Originally rated prior to the label being introduced but resubmitted after the label was introduced for another rating (might be an updated version, part of a new compilation, or a new hardware platform release) | *Rainbow Six Siege* | PEGI | duly labelled | recognizes had opportunity after the label was introduced to re-label the game upon resubmission **admits to failing** |
| | | resubmission for new hardware platform releases (made after the label was introduced) received the label | Commits to improving procedure for similar resubmissions in the future |
| | | Original submission remains unlabelled | |
| *Apex Legends et al.* | ESRB | not labelled | duly labelled |
| *Black Desert et al.* | | no intention to add the label in the future | invited publishers to add the label |
| *Hunt: Showdown et al.* | | failed to review rating when given opportunity after the label was introduced upon resubmission: 're-releases of games that do not add new pertinent content use the initial submission's rating assignment'. | relevant publisher voluntarily added the label upon invitation |
| *Genshin Impact* | ESRB | **admits to labelling error** | duly labelled |

announcements, although applying the label retroactively should be preferred. This incongruence with retroactivity affects the fair interpretation of their label application's 'consistency rate' (which has been considerably reduced as a result) when calculated as preregistered. A more equitable consistency rate must be determined that accounts for the retroactivity issue because the ESRB has been attributed with a substantially higher number of failings than may be fair.

I have since received official responses from PEGI and the ESRB explaining the 40 disagreements/inconsistencies: these are synthesized in table 1. Prior to receiving those responses, I conducted exploratory analysis attempting to deduce the explanations for the inconsistencies: that content remains available as §3.1.2 of the first draft of this study: https://doi.org/10.31219/osf.io/asbcg (Version 1 dated 18 January 2023). I do not expect that the PEGI and the ESRB explanations would be intentionally incorrect and misleading as both were aware that their responses would be duly published at the data deposit link. The official explanations are more reliable than, and preferable to, my own deductions, and thus are presented instead.

The official responses allowed for the retroactivity issue to be dealt with fairly and accurately. In total, 30 games (29 games that the ESRB did not label and *Mobile Suit Gundam Battle Operation 2*, which PEGI did not label) fall within this category of games that were rated prior to the introduction of the label and did not have another variation of the same title resubmitted (for whatever reasons) for a separate rating after the label was introduced, which would give the age rating organization another fresh opportunity to reconsider whether to attach the label. Four games were confirmed not to contain loot boxes by at least one age rating organization and were either confirmed or likely false positives at the other. One game took technical measures, specifically, geo-blocking, to prevent loot boxes from being accessible in PEGI territories, thus rendering the version of the game rated by PEGI to not be the same as the one rated by the ESRB. For the justifications given, all 35 aforementioned games must be excluded from the revised consistency rate calculation.

This leaves five games, which in the author's view must be deemed as inconsistent due to genuine, culpable failings on the part of either the ESRB (four games; 80%) or PEGI (one game; 20%). The ESRB admitted fault for only one of those four games, while PEGI admitted fault for the one single game. The one fault that the ESRB admitted to as a rating mistake is not worth further elaboration. PEGI admitted fault with one game even though this game was originally rated before the label was introduced, because this game has since been resubmitted at a date after the label came into force as part of a compilation, thus giving PEGI a genuine opportunity to attach the label after it was introduced, and PEGI failed to add the label at this new opportunity. The three games for which the ESRB would not admit fault fall within the same category that PEGI admitted fault for: the ESRB had a new opportunity after the label came into force to re-examine each of those games for labelling due to a resubmission (although the ESRB disputes whether these should technically be viewed as 'resubmissions' *per se*). However, the ESRB failed to use these opportunities to consider whether labelling was appropriate and simply copied the initial submission's rating (even though that rating was decided using outdated criteria). Notably, the ESRB did properly label *Rainbow Six Siege* (an older game originally rated prior to the label's introduction) upon resubmission for new hardware platform releases after the label was introduced, so there was even 'precedent' of this being done correctly at the ESRB.

Among a total of 31 games that the ESRB and PEGI both had fair opportunities to label after the label was introduced: 26 games were consistent, while five were inconsistent due to one organization culpably failing to label the game. The revised 'consistency rate' is 83.9% among this subsample.

Hypothesis 1 would still be rejected even when the more equitably determined consistency rate of 83.9% is used, as it remains less than 95%.

## 3.2. Study 2: IARC on the Google Play Store

### 3.2.1. Confirmatory analysis

From the list of 431 games derived as preregistered, 129 games previously assessed to have *not* contained loot boxes (potentially including duplicate games) were removed, resulting in a list of 302 games. From that list of 302 games, 127 duplicates, *Roblox* and *Minecraft* were removed, thus leaving a list of 173 non-duplicate games previously assessed to have contained loot boxes.

When forming the sample of 100 random games previously known to contain loot boxes, six games that were initially randomly chosen were excluded and replaced. *Guns of Glory* was replaced as preregistered. *Brawl Stars* (Supercell, 2017) and *Mario Kart Tour* (Nintendo, 2019) were replaced because these two games were known to have had their loot boxes removed since 18 September 2022

and therefore no longer contained loot boxes [60,61]. Interestingly, *Brawl Stars* still had the label attached as of 12 January 2023, despite not containing any loot boxes, and thus it would have technically constituted a false positive. By contrast, *Mario Kart Tour* did not have the label attached as of 13 January 2023, and it could not be determined whether a label was duly attached *previously* when the game did still contain loot boxes. *Hokage Ninja Duel* (unknown developer/publisher, unknown year) was not available on the Google Play Store as of 13 January 2023 and was replaced. *Clawee* (Gigantic, 2017) was available on the Google Play Store, but the game would not operate on the phone after being downloaded despite multiple attempts at different times and was replaced. *Sniper 3D Assassin: Gun Games* (unknown developer/publisher, unknown year) could not be found under this exact name on the Google Play Store. A game entitled '*Sniper 3D: Gun Shooting Games* [the Chinese full-width colon is original]' (Fun Games For Free, 2014) was identified as a potential match; however, although said game was available when browsed using the web version of the Google Play Store, it could not be found on the phone version of the Google Play Store. Downloading and installing the relevant apk (Android Package) file from an alternative source was deemed to be potential copyright infringement and ill-advised. Given that the game is not even a confirmed match (as no information on which company was operating this game when Zendle *et al.* [34] originally coded it is available), it was deemed more appropriate to exclude and replace this game.

Among the 100 random games previously known to contain loot boxes which were sampled, 29 games (29.0%) displayed the label and were therefore assumed to continue to contain loot boxes and 71 (71.0%) did not display the label. The Danish and the US storefronts were identical as to whether a game was labelled, thus indicating that the system did indeed generate the PEGI-worded and ESRB-worded ratings from the same source information (i.e. the developer/publisher questionnaire [62,63]). Of the 71 games that did not display the label and were therefore replayed to verify whether they continued to contain loot boxes, 71 games (100.0%) were found to continue to contain loot boxes. The 'compliance rate' among the entire sample was therefore calculated to be 29.0%.

Hypothesis 2 is rejected as the compliance rate of 29.0% is less than 95%.

As preregistered, 13 games were identified to be 'social casino' games and were deemed to 'contain loot boxes', among which two games were labelled with the 'loot box presence' warning meaning that the compliance rate was 15.4%. This shows that some companies (although this appeared to be a minority view) do recognize social casino games as broadly falling within the 'loot box' or the 'in-game transactions with randomized elements' [3] definition. Among 87 non-social casino games, 27 were labelled, therefore giving a 31.0% compliance rate. Notably, many social casino games received PEGI 12 and ESRB Teen ('suitable for ages 13 and up' [28]) ratings. Two such games even received PEGI 3 and ESRB Everyone ('suitable for all ages' [28]) ratings. Whether those age ratings are appropriate is left to the discretion of the relevant stakeholders. However, PEGI has previously decided that, since 2020, all new games that contain simulated or actual gambling are to be rated PEGI 18 [27]: this is demonstrated by how social casino games frequently received the arguably incongruent combination of PEGI 18 and ESRB Teen ratings.

Neither *Roblox* nor *Minecraft* displayed the loot box presence warning label on the Google Play Store when assessed on 12 January 2023.

### 3.2.2. Exploratory analysis

I have since received official responses from PEGI and the ESRB/the IARC explaining the actions that they have taken in relation to the non-labelled Google Play Store games. Notably, the responses from the ESRB and the IARC were received from one relevant person who acted in her capacity as both President of the ESRB and Chairman of the IARC. For this reason, at times, it was difficult to discern whether a response should be properly attributed to the ESRB, the IARC, or both. Prior to receiving those responses, in addition to the 71 games identified as non-compliant through confirmatory analysis, I conducted exploratory analysis, justified in §3.1.2 of the first draft of this study: https://doi.org/10.31219/osf.io/asbcg (Version 1 dated 18 January 2023), that found a further 13 non-compliant games. I sent this list of 84 games to the age rating organizations asking for their response and, if appropriate, corrections.

The IARC response may be summarized as follows. The label was only introduced to the Google Play Store in February 2022 (i.e. almost 2 years after it was introduced for physical games, but nearly a year before the data collection dates). The IARC opines that any games originally rated prior to February 2022 need not be attached with the label and should not be deemed as non-compliant. The IARC admits fault as to three games 'legitimately missing' the label. The IARC's response suggested that the measure was

well complied with after February 2022. I found this suggestion unconvincing: I address this in §4.2.1, but I also undertook further exploratory analysis to check the IARC suggestion that supposedly the label was well complied with by games rated/released after February 2022. Specifically, with little effort, I found an additional list of 11 games released after February 2022 that most likely contained loot boxes and were not labelled. This list includes *Diablo Immortal* (Blizzard Entertainment & NetEase, 2022) whose loot boxes were highly controversial and publicized due to Blizzard Entertainment deciding not to release the game in The Netherlands and Belgium, citing gambling regulation ([35], p. 13). The other 10 games were recently released so-called 'gacha' games. Further data collection was stopped because the author deemed the point proven that many games released after February 2022 were still not duly labelled. The IARC response in relation to these 11 games was that 'a large portion of them were submitted [for rating] prior to [the label being introduced]'. The rating submission dates of games cannot be independently verified.

I was informed by the IARC on 31 January 2023 that it deemed five of those 84 games to '*not* warrant the [label]' (emphasis original) and that all the other games (regardless of when they were originally assigned a rating) on the list of 84 games were added with the label. The divergent interpretations as to whether a game warrants the label are discussed at §4.2.2. I since checked whether those other 79 games were duly labelled and, as of 5 February 2023, only 77 games were so labelled. Further communications with the IARC revealed that two games were marketed as multiple products in different regions, rather than one product in all regions. The IARC only labelled the North American versions but not the European or 'Rest of the World' versions. The versions of the two games that I originally found and later identified to the IARC have since been labelled; however, one additional 'Rest of the World' version of one of those two games remained unlabelled as of 11 February 2023 because I did not specifically highlight it to the IARC. Ten of the games on the list of 11 games released after February 2022 that were unlabelled when I first checked were also since labelled through rectification as of 7 February 2023, thus reflecting that *at least* all but one (as no explanation has been provided for the unlabelled game) did contain loot boxes despite the author not having personally confirmed that point through gameplay.

The IARC refused to label *Roblox* and *Minecraft* as detailed under §4.2.4.

# 4. Discussion

At the outset, I note that I have since had communications with PEGI and the ESRB/the IARC following data collection. All such communications (except for a remote meeting with PEGI) were in writing and are published at the data deposit link for transparency and further scrutiny. I endeavour to provide my interpretation of the present findings (informed by PEGI's and the ESRB's/the IARC's responses) below. From my experience, PEGI has been more forthcoming in admitting to problems and willing to consider how to fix those issues in the future. By contrast, I have found the response from the ESRB/the IARC to be defensive and unpersuasive: it did not show a willingness to admit to problems or a desire to improve in the future. My opinion is reflected below. An initial draft and non-peer-reviewed version of the Results, Discussion and Conclusion sections written prior to any communications with the relevant organizations is available at: https://doi.org/10.31219/osf.io/asbcg (Version 1 dated 18 January 2023). I welcome others to come to their own conclusions based on the present findings and the responses from PEGI and the ESRB/the IARC.

## 4.1. Study 1: physical games: the ESRB and PEGI

### 4.1.1. Retroactivity

The consistency rate of the ESRB's and PEGI's application of the loot box warning self-regulation was 39.4% in the overall sample. This disappointingly low result is somewhat misleading (and arguably treated the ESRB unfairly) due to the ESRB's and PEGI's inconsistent approach to the measure's retroactivity. The ESRB's announcement of the loot box warning label was not clear as to whether it would be applied retroactively to games that have already been rated [3]; similarly PEGI's announcement of its label was also not clear about retroactivity [24]. One could have fairly assumed from the language used by both (specifically, the future tense) that retroactivity was not originally intended by either. The ESRB stated: 'This new Interactive Element … *will be assigned* …' [3] (emphasis added), while PEGI stated: ' … publishers *will start to provide* additional information …' [24] (emphasis added).

The ESRB and PEGI provided official responses as to retroactivity. The ESRB has *not* applied the label to games retroactively and has refused to change this approach when responding to the present study (as detailed below), while PEGI has invited publishers to *voluntarily* apply the label (but did not *require* them to do so) by sending communications twice a year. Study 1 could not determine whether all historical games that contained loot boxes have been accurately retroactively labelled by PEGI, at least one instance (*Mobile Suit Gundam Battle Operation 2*) where this was probably not done was detected. PEGI cannot promise that the label's retroactive application to historical games has been complete and accurate, but it should be recognized for taking proactive action to do this: most, if not all, well-known historical games containing loot boxes have now probably been duly labelled.

The two age rating organizations have taken divergent actions in relation to the same consumer protection measure. Undoubtedly, PEGI's approach of applying the label retroactively should be recognized as being better for consumers because more information has been provided even in relation to older games. (An 'older' or 'historical' game in the ESRB/PEGI context is defined as a game originally rated prior to the announcement date of the loot box warning label of 13 April 2020, which means that some of these games were released less than 3 years ago and *not* many years ago.) One can appreciate that providing this label for older games that few players are likely to play today (e.g. an outdated version of the *FIFA* games), and which might not even be in operation any more (e.g. the servers might have already been shut down, thus rendering loot box purchasing factually impossible), may lead to wasted costs. However, given that very little costs would be expended by doing so, retroactivity appears justified to minimize potential loot box harms. PEGI achieved this with little effort simply by sending communications to all publishers twice a year. Any companies that are continuing to offer loot boxes in older games should know that they are doing so and can be expected to easily report this to the ESRB and PEGI and get the label attached at minimal costs. Notably, the German age rating organization, the USK, announced on 14 December 2022 that it will also begin to attach loot box presence warning labels to 'newly submitted' games from 1 January 2023 [38]. The USK has thereby avowed that it will *not* apply the label retroactively.

Going forward, given the consumer protection benefits and minimal compliance costs of adopting retroactivity, the ESRB and the USK should emulate PEGI's example and apply the labels to previously rated games. This process could also be made more efficient if the age rating organizations work collaboratively: for example, the ESRB and the USK should immediately label any historical games that PEGI has already since labelled (or at least promptly seek a clarification from the relevant game company and make a decision accordingly). All three age rating organizations should also update their policy and *require* relevant companies of all older rated games to provide an update as to loot box presence if this is relevant, rather than merely *recommending* companies to voluntarily do so, which is PEGI's current approach. If no response is received from the relevant companies of the older games after a reasonable period of time (e.g. a month), then a warning message stating that the loot box presence status of said game could not be determined should be appended to the online age rating to inform consumers accordingly and pressure the company into replying. Such a message would help to avoid misleading consumers into thinking that a game does not contain loot boxes (which is what the lack of a label presently incorrectly implies). PEGI responded to this suggestion with the counter-argument that this might mean that many games not containing loot boxes are uselessly attached with this message. Parents and consumers may become desensitized to loot box-related warnings as a result. That makes sense; however, ideally, most companies would reply promptly so this would not occur.

A centralized resource should also be developed: if any one rating system decides to label a historical game or indeed a new game as containing loot boxes, that decision should be communicated with others to ensure consistency across the various rating systems. Such cooperation can act as a safeguard to check each other's mistakes and omissions and enhance the accuracy of the labelling process. This type of cross-checking is sensible in relation to the loot box warning label specifically because whether a game contains loot boxes is not (or at least should not be) subjective, unlike, for example, the amount of violence and sexual content that different cultures would allow for various age groups (which various age rating organizations may, and indeed do, reasonably disagree about).

However, cross-checking also has its limitations, so rating systems should encourage players and other stakeholders (such as parents and other competing companies) to report non-compliance (specifically, missing labels). This can help to address cases where all rating systems have failed to notice the loot box implementation in a certain game (which is not unimaginable, as this would probably be due to the relevant company submitting the same inaccurate information to all rating systems). Such complaint avenues were already open prior to this study. However, the ESRB reported that no complaint has ever been filed in relation to inaccurate labelling. Given that a highly popular

game (*Genshin Impact*) was mistakenly left unlabelled (as detailed below), this suggests that the complaint system should be more widely advertised and that accurate complaints leading to the identification of a genuine error should perhaps be financially incentivized to encourage more active reporting. When a mistake or omission has been identified, rating systems should proactively pursue enforcement action (e.g. immediate rectification and fines of 'up to [US]\$1 million' [63] according to the ESRB or 'up to €500 000' [64] according to PEGI, as appropriate).

Unfortunately, the ESRB has refused to adopt the aforementioned suggestion of applying the label retroactively without providing a justification (beyond that this was not originally planned). This is despite PEGI having admirably already largely successfully accomplished this and the fact that doing so would only incur minimal costs. Indeed, even if a company voluntarily wanted to add the label, presently, it would not be allowed to change the rating information in the ESRB database found through the online search tool, although the ESRB has 'encouraged' companies to add the label to physical 'packaging and marketing materials'. A major publisher deciding to act responsibly could probably pressure the ESRB into changing its system and allowing labels to be retroactively attached, but no publisher has yet done this (even though they have voluntarily done this at PEGI). Given the ESRB's refusal to improve its approach, it must be concluded that the ESRB 'In-Game Purchases (Includes Random Items)' label was not complied with and was not enforced to a satisfactory degree. Parents and all stakeholders should place cautious reliance on this measure. Games marked with the label should be treated as containing loot boxes and due caution should be exercised (although a few, probably harmless, instances of false positives were also identified). However, some non-labelled games still contain loot boxes, so all non-labelled games should not be assumed to not contain loot boxes. In particular, consumers (including parents) living in ESRB territories are well advised to additionally refer to the PEGI rating system for more complete and accurate information on loot box presence as many older games containing loot boxes remain unlabelled by the ESRB. This burden to cross-check should *not* be unfairly placed on consumers; however, given the ESRB's refusal to adopt the recommendations above and improve its labelling's accuracy and reliability, ESRB consumers must proceed with due caution and not place over-reliance on this questionably discretionary measure.

By contrast, PEGI's implementation of the 'In-Game Purchases (Includes Random Items)' label is more satisfactory. PEGI should be commended for retroactively applying the label and also for committing to improving its rating procedure in light of the one identified and admitted mistake. The PEGI measure could reasonably be relied upon by parents to provide accurate information (barring one or two mistakes and false positives, which PEGI has demonstrated that it is willing and able to promptly correct).

Indeed, leaving aside the ESRB's unjustifiable refusal to apply the label retroactively and to re-releases, with newly released games (i.e. not re-releases), consumers can rely on the PEGI and ESRB labels, although the rating procedures can still be improved.

### 4.1.2. Non-compliance

After addressing the retroactivity issue in an equitable manner and duly accounting for PEGI's and the ESRB's official responses, the revised 'consistency rate' becomes 83.9%. Five cases of culpable failings (16.1%) remain, which is quite unsatisfactory. As preregistered, as this consistency rate falls between 80% and 95%, it must be interpreted as the self-regulatory measure not having been applied sufficiently consistently by the ESRB and PEGI, and thus the rating processes require improvements to enhance cohesion. Further regulatory issues are illustrated through a case study on a non-compliant game: *Genshin Impact* (miHoYo, 2020), which was attached with the label by PEGI but *not* by the ESRB. For context, *Genshin Impact* was attached with the label by the IARC on the Google Play Store as of 13 January 2023. The ESRB has admitted fault for failing to attach the label to *Genshin Impact* and, as of 7 February 2023, has corrected the rating and added the label.

The ESRB claims in relation to 'Physical Games' it has rated that: 'After release, testers may also play-test the game to verify that the content disclosure was complete' [63]. It is not known whether *Genshin Impact* was in fact physically released as it has been a predominantly digitally released game. However, miHoYo, the developer and publisher, specifically applied for a rating from the ESRB (rather than merely relying on the IARC, which would have sufficed for digital release) meaning that physical release was at least contemplated. However, the ESRB does not appear to have subsequently asked testers to verify whether *Genshin Impact* contained loot boxes and therefore should have the label attached. If the ESRB did invite scrutiny following the release of *Genshin Impact*, the testers failed to identify loot box presence and made obvious errors in their judgement, as the gambling-like, loot box (gacha) character summoning mechanic of *Genshin Impact* is prominently marketed and central to the player's experience and gameplay progression

([65], p. 1077); has been available since the release of the game (and so the ESRB could not have rated an older version without the loot box mechanic); has been correctly identified in two previous loot box prevalence studies [18,35]; and has been subject to media reporting and criticism in popular channels (e.g. [66]).

The ESRB response as to testing after release is that this process was done, but it was 'largely focused on ensuring that all pertinent content that might impact the assignment of an age rating or content descriptor was disclosed during the rating process'. The ESRB decided that the loot box presence warning should not be a 'content descriptor' (e.g. 'Use of Drugs', which would influence the age rating) and should instead be an 'interactive element' (which would not influence the age rating). However, whether the label is one artificial category of content or the other should not affect whether it is duly checked during post-release testing. It is arbitrary to 'largely focus' post-release testing on 'content descriptors'. Indeed, many parents would probably be more concerned about the 'Users Interact' (which includes user-to-user communications); the 'Shares Location' (which includes features that allow other users to see a child's real-life geographical location); and the 'Unrestricted Internet' (which, as the name suggests, allows access to any website) interactive elements than they are of even the most concerning 'Content Descriptors', as the former are capable of posing much more direct harm to a child, including physical injury. The ESRB's post-release testing process is inadequate and must be improved to always account for 'interactive elements'. This official response also exposes potential issues with wider child protection that the ESRB may not be equipped to address.

The rating error of *Genshin Impact* is due to miHoYo (the developer and publisher) failing to submit relevant information and the ESRB both failing to attach the label initially when reviewing miHoYo's submission and then subsequently failing to rectify its mistake. The ESRB rating system failed to catch this error at multiple stages. It is highly unsatisfactory that such a game did not have the loot box warning label attached for more than 2 years after release. *Genshin Impact* is a prominent example of a free-to-play game that operates using the game-as-a-service business model. Many such games are monetized predominantly based on a gambling-like loot box system and are regularly updated with new content (which is often made available in the form of highly desirable, new loot boxes rewards that may be stronger than those obtained through other means ([30], p. 182), e.g. new playable characters in *Genshin Impact* that are only obtainable through the above-mentioned character summoning system). Since its release, *Genshin Impact* has been, and continues to be, one of the most popular and highest-grossing video games, as demonstrated by its repeated nomination for 'best mobile game' and 'best ongoing game' (in the 2 years subsequent to its initial release) at The Game Awards, arguably the leading video game awards ceremony, in 2020 [67], 2021 [68] and 2022 [69] and for winning the Player's Voice category, which is entirely decided by the general public, in 2022 [70]. In relation to such 'ongoing' games with frequent updates, it is important that compliance with all consumer protection measures is also kept up to date. This represents another important reason why loot box warning labels should be applied retroactively. In particular, non-compliance by older games that are being regularly updated and are still high grossing should not be tolerated.

### 4.1.3. Rating process lacks accountability

In relation to three games: *Apex Legends* (Respawn, 2019), *Black Desert* (Pearl Abyss, 2015) and *Hunt: Showdown* (Crytek, 2018), which were originally rated by the ESRB prior to the label coming into non-retroactive effect, the ESRB had more recent opportunities after the label came into effect to revisit its rating decisions when rating downloadable content (DLCs), newer editions or releases on newer platforms of these games. However, it has failed to update the rating and attach the label to the newer versions of these games and admitted to having simply relied on and copied the older decision. The relevant game companies may also be to blame for having submitted outdated and inaccurate information and failing to disclose or specifically highlight loot box presence to the ESRB in their more recent submissions. As a counter-example, *Tom Clancy's Rainbow Six Siege* (Ubisoft, 2015) was originally not labelled by the ESRB; however, the newest hardware platform releases were duly labelled. This may be due to Ubisoft submitting updated information correctly highlighting loot box presence.

The rating processes at the ESRB [63] and PEGI [62] are very similar. There are basically two steps. Firstly, the publisher completes a questionnaire which asks for the self-disclosure of relevant content. The ESRB states that its questionnaire asks for details on 'other factors such as … reward systems … ' [63] This probably refers to loot box presence. It can be assumed that the PEGI questionnaire also asks about loot box presence. The publisher then submits either the game content (at PEGI [62]) or a video showing all relevant game content (at the ESRB [63]) for review by the relevant age rating organization. The ESRB

promises that 'at least three trained raters' would recommend, *inter alia*, 'Interactive Elements' (which includes the 'In-Game Purchases (Includes Random Items)' label) [63]. Further, the ESRB promises that 'After release, testers may also play-test the game to verify that the content disclosure was complete' [63]. PEGI promises that 'PEGI administrators … thoroughly review the provisional age rating [which is automatically derived from the self-disclosure questionnaire]' [62]. No information is provided on post-release monitoring by PEGI. PEGI has shared that loot boxes are particularly difficult to rate because the version of the game it playtests would not be the official release with servers connected and so loot boxes may not appear in the in-game shop that PEGI observes. Therefore, heavy reliance must be placed on self-disclosure during the first step.

This rating process is advertised as being rigorous; however, scrutiny by the present study revealed that (even accounting for the retroactivity issue in a most generous manner) at least 16.1% of games containing loot boxes were not duly labelled by one of either age rating organization. There may also be games that were labelled by neither organization. It also could not be determined what percentage of games already had the loot box presence warning label attached after the questionnaire was filled in by the publisher (i.e. after self-disclosure) and what percentage did only after review of the submission by the age rating organizations (which would mean that the self-disclosure was inaccurate). Both age rating organizations should review their records to figure this out. Doing so would help to identify whether the first step of the rating process needs improvement: if most games did not self-disclose and only had the label attached in the second step upon review by the organizations, then the organizations should consider conducting education campaigns for the publishers to improve the accuracy of the self-disclosures during the first step. The organizations should also compare their records as this could help to identify whether the PEGI approach of reviewing the actual game content might be a superior approach that helped to identify more loot boxes during the second step than the ESRB approach of reviewing only a video of the game content (e.g. in relation to how the ESRB failed to label *Genshin Impact*, but how PEGI did so).

Both age rating organizations responded promptly to my complaint and investigated the relevant issues, including making corrections (although the ESRB has not admitted fault as willingly as PEGI did). Both organizations already provided a complaint system prior to this study, and I hope both would keep such channels open and also proactively respond to complaints filed by others, including parents, and, if appropriate, revise the ratings and labelling status. This is probably the easiest and most cost-effective way of ensuring that information is kept updated and accurate. Two improvements to the rating process are also recommended. Firstly, whenever a newer release of an older game (e.g. DLCs, new editions or new platform releases) is to be rated, the rating process should start afresh; not rely on older ratings; and ensure that any updated content and updated rating guidelines (e.g. to label loot box presence) are duly accounted for. PEGI has committed to improving its rating procedure to more accurately address older games containing loot boxes that get resubmitted. By contrast, the ESRB has refused to adopt this recommendation, meaning that older games would still not be labelled upon re-release. Secondly, due to how contemporary games are frequently updated either to add or remove loot box content, it appears fair to ask for the questionnaire to be refilled and the rating process reconducted annually or at least upon a major content update. This would lead to additional compliance costs, but these costs would be very minor in the context of other regulatory requirements generally (*inter alia*, annual financial reporting obligations), and is justifiable given the potential harms of loot boxes and the need to ensure that any measure that is promised to consumers is complied with effectively and accurately.

### 4.1.4. Enforcement?

The ESRB promises that 'we have several mechanisms in place to ensure that publishers fully disclose all the content in their games, so consumers get complete and reliable rating information' [63]. physical games, which are relevant to Study 1, the ESRB states that 'our enforcement system includes sanctions and fines (up to $1 million) that may be imposed on publishers who don't fully disclose content to us during the rating process' [63]. Similarly, violations of the *PEGI Code of Conduct*, including failure to disclose significant content (which undoubtedly includes loot box presence), may be sanctioned with 'fines of up to €500 000' [64]. Legal sanctions may also apply to such violations of industry self-regulation, e.g. criminal breaches of the EU Unfair Commercial Practices Directive [2005] OJ L149/22, annex I, para 4. The suspected non-compliant games were, respectively, referred to the ESRB and PEGI, who were asked to take enforcement actions against the relevant companies. The ESRB has corrected its error with *Genshin Impact*, and PEGI has promised to improve its procedure for re-assessing resubmitted older games. However, it is not known whether the relevant companies involved would be punished with a fine.

### 4.1.5. Scale of the loot box issue: platform differences

Finally, if we are to trust the ESRB and PEGI that only 74 games, among all physical games ever rated, should have had the label attached and therefore contained loot boxes, this reflects that the scale of the loot box and predatory monetization problem is very different on the console/PC platforms as compared with the mobile platform. Most loot box prevalence studies on the mobile platforms identified as many games (usually more) as to have contained loot boxes simply by examining the 100 highest-grossing games [18,20,34,35]. PEGI told me in a remote meeting that only 2%–3% of all console and PC games it rates contain loot boxes. Therefore, in terms of the real-world implications as to the number of games concerned, only a minority of loot box-containing games would fall within the direct jurisdiction of the ESRB and PEGI. (However, it is important to note that some of the games in this seemingly small proportion of games containing loot boxes, such as the *FIFA* game series, do generate a disproportionately large amount of revenue; are played by many players; and are of more general significance than many other games that do not contain loot boxes.) Therefore, whether the IARC is attaching the label accurately is probably of more practical importance. This is addressed in Study 2.

## 4.2. Study 2: digital games: IARC on the Google Play Store

### 4.2.1. Non-compliance

Only 29 of the 100 random games previously known to contain loot boxes attached the loot box presence warning label on the Google Play Store through the IARC system. The other 71 games were replayed to confirm that all of them did indeed continue to contain loot boxes as of mid-January 2023 and therefore genuinely failed to disclose loot box presence. The non-compliance rate was 71.0%.

At present, the IARC 'In-Game Purchases (Includes Random Items)' label cannot be treated as a trustworthy and dependable authentication of whether a game contains loot boxes or not on the Google Play Store. Parents and all stakeholders should proceed on the assumption that a game marked with the label does indeed contain loot boxes (although, again, note that at least one, probably harmless, false positive was observed), but no reliance should be placed on the lack of a label, because a non-labelled game may still contain loot boxes. Indeed, unlabelled games that are high grossing are quite likely to still contain loot boxes, given the results of previous loot box prevalence studies [18,20,34,35]. Advertising this measure as providing consumer protection when it has been poorly complied with in practice gave consumers a false sense of security. Further elaboration on this negative consequence has been made elsewhere in relation to poor compliance with Apple's loot box probability disclosure requirement [18] and Belgium's failed loot box 'ban' [35], and is therefore not repeated.

Given this very poor compliance rate, one must question whether this measure was another disingenuous, perfunctory attempt by the industry to dissuade legal regulation (similar to poorly complied-with industry self-regulation requiring loot box probability disclosures [18]). The IARC's defensive responses that fail to suggest tangible ways of improving this currently unsatisfactory situation (detailed below) support such an interpretation. Unless the measure is significantly improved, and until it is independently tested again and found to be effectively complied with, policymakers should place very little (if any) reliance on this measure when determining loot box regulation in the context of mobile games (which does appear to be the crux of the issue) going forward.

The official IARC response has been that the label only became effective on the Google Play Store in February 2022, and therefore, in its opinion, any games originally rated prior to February 2022 need not be labelled and should not be deemed as non-compliant even if unlabelled. This is an important disclaimer that was not publicly known until the present study was conducted and still has not been prominently published (e.g. alongside the IARC page promising the implementation of this measure and on the Google Play Store page concerning ratings). I also find this excuse unconvincing. Firstly, I have demonstrated that many games released *after* February 2022 were still unlabelled (although the IARC insisted that many of those games were, in fact, rated before February 2022; however, this cannot be independently verified). This exploratory analysis bolsters the sample used for the confirmatory analysis, which was derived from historical samples of games known to contain loot boxes and were, therefore, invariably older games. (An 'older' or 'historical' game in the Google Play Store IARC context is defined as a game rated prior to February 2022. Many games are released months after their rating date, which means that many of these so-called 'older' games were released just a few months ago and are not 'old' in a practical sense.) The IARC has revealed that, even for newly released games, the measure still has not yet fully come into force many months after its

introduction to the Google Play Store. Put another way, there is no specific deadline for compliance. This must now be required to ensure accountability. Secondly, every single one of the 29 *compliant* games were released/rated before February 2022, and each of those games managed to correctly attach the label to itself despite being an older game. Indeed, because some older games were labelled and compliant, many consumers would be under the false impression, and therefore expect, that all games would be accurately labelled. Thirdly, it is illogical and unjustifiable for a consumer protection measure like this to not apply retroactively to older games that continue to be in operation, particularly considering that every single one of these games received a software update *after* February 2022 (which represented opportunities for the IARC to force them to complete a new questionnaire asking about loot box presence). As the IARC system currently stands, a game originally rated before February 2022 that later adds loot boxes after February 2022 would also not need to attach the label. Older games have a blanket licence to implement loot boxes with impunity. Just because a product was originally released prior to a consumer protection measure coming into force cannot justify newly produced versions of that product to not incorporate a consumer protection measure (particularly considering the ease and minimal costs with which this measure can be implemented and complied with: the publisher only needs to edit their answers to one item in the questionnaire).

To use an analogy to illustrate the IARC's 'justification': a consumer protection measure now requires all hamburgers to disclose whether they contain a certain ingredient, but one restaurant argues that it need not comply with this measure because it has been producing hamburgers before this measure came into force. No one would accept this restaurant's argument. The legal principle of non-retroactivity applies only to acts done entirely before the measure came into force and *cannot* apply to acts that started being done before *but could have since been stopped* but have not been stopped without a proper justification. In theory, only a game that was rated and released prior to February 2022 and is no longer being operated need not attach the label: this would require this game to no longer sell loot boxes and be removed from the Google Play Store, as no party should directly benefit from that game's operation any more. Any games that remain on the Google Play Store that continue to operate, sell loot boxes, and generate revenue past February 2022 must be liable for labelling.

The IARC has attempted to 'shift the goalpost' as to what is 'compliant'. However, whether a game is 'compliant' should be based on the view of the reasonable person, and the IARC cannot redefine that for its own benefit: a parent would naturally expect this measure to be applied consistently across all games regardless of any game's original rating submission date if that game continues to sell loot boxes. The original rating submission date that the IARC relies on is also information that could not be independently verified, and thus including this as a criterion for 'compliance' would also mean that this measure cannot be subjected to independent, external scrutiny. A self-regulatory measure cannot be without accountability.

As detailed in §4.1.3 above, the ESRB and PEGI have a two-step process for rating physical games: the self-disclosure by companies is reviewed by the two organizations with additional material. In the IARC context, there is only one step: self-disclosure by companies. Many companies failed to accurately self-disclose and attach the label. It is not known whether the IARC or Google undertook to inform companies about the newly introduced label and encourage them to attach it by completing an updated version of the questionnaire, although a minority of companies did do so. In Study 1, it could not be ruled out that there were games containing loot boxes that neither the ESRB nor PEGI managed to identify and so a 'compliance rate' with the ESRB/PEGI measure cannot be determined for direct comparison purposes. However, it is likely that the two-step process was more effective at identifying loot boxes, specifically, that the second step of external review helped to identify additional loot boxes (although PEGI has stated that these often could not be directly perceived in playtest versions that it rates). The IARC was established because, as PEGI claims, 'This traditional method [of pre-release, two-step review] is not at all practicable for purely digital storefronts that see thousands of new products enter the market (and even more products updated or changed) on a daily basis' [62]. However, Study 2 has evidently shown that the IARC is not working effectively in relation to self-disclosing loot box presence. The adoption of a pre-release, two-step review process for *all* digital games might not be practicable, but an adapted, post-release form of two-step review (that age rating organizations have direct external oversight on) could potentially be required. For example, the highest-grossing games should be reviewed every three months to ensure sufficient scrutiny of at least those games. The relevant companies profiting from loot boxes could be made to bear the associated increased compliance costs. The IARC has responded stating that post-release monitoring does take place on the Google Play Store; however, this has been limited only to ensuring that the game complied with the rating guidelines that were in force upon its submission. In other words,

older and newer games are monitored with a double standard. Anti-competition authorities might be interested in considering the antitrust implications of regulating older video games (and, by implication, more established companies) with fewer rules, while forcing newer games (many of which would be released by smaller, emerging competitors) to abide by more stringent requirements.

### 4.2.2. Corrections by the IARC since data collection

I submitted a list of 84 unlabelled games to the IARC asking for rectification as detailed in §3.2.2. The IARC raised a number of counter-arguments against labelling games that were originally rated prior to the label coming into force on the Google Play Store. One argument that I found sensible (although not wholly convincing) was that: only the games I identified would be labelled, while many more other games containing loot boxes that I did not examine would not be labelled, and the fact that this may be perceived as unfair by companies. However, the IARC decided to label all of the games from that list as long as it was re-verified by the IARC to contain loot boxes. After conducting a review, the IARC promised that 79 of those games would be labelled, but, as of 5 February 2023, only 77 of those games were confirmed to have been labelled, and seven games were left unlabelled. Further communications resolved the issue with the two then unlabelled games amongst the 79 games as detailed in §3.2.2. However, the IARC's failure to label other release versions of the same game demonstrates that it lacks broad control over the Google Play Store and can only act on a case-by-case basis. An extreme example of a game, *Castle Clash* (IGG, 2013), that is currently released as 15 separate Google Play Store entries is shown in figure 4. I reported the English version to the IARC, which has since been labelled, but, as of 9 February 2023, none of the other versions were labelled even though they were collectively downloaded over 35 000 000 times.

The IARC determined that five of those 84 games I reported do not contain loot boxes and therefore do not warrant the label. After initially refusing to do so, the IARC was eventually convinced by me to share the titles of the five games that allegedly do not warrant the label. The IARC examined a different game for one of those five games. I concede that one of the other four cases might be debatable: the Pokémon egg and incubator mechanic in *Pokémon Go* (Niantic, 2016) has been inconsistently treated in the academic literature as to whether it constitutes a loot box ([56]; [71], p. 26) or not (cf. [34,57]). However, the other three games contained obvious and uncontroversial loot box mechanics (whose screenshots are available at the data deposit link). *Football Rivals: Online Soccer* (Green Horse Games, 2020) is a social casino game involving a slot machine and an energy system used to activate that slot machine. Energy can be refilled with real-world money. A similar social casino game example, *Board Kings: Board Dice Games* (2017, Playtika), involving dice and an energy system, was duly rectified and since labelled. *Harry Potter: Puzzles & Spells* (Zynga, 2020) contained classic loot boxes that became available for perusal after a considerable (but still less than one hour) length of gameplay. The loot boxes in *Final Fantasy XV: A New Empire* (Machine Zone, 2017) were available as part of larger bundles and therefore more difficult to identify. The IARC playtesters probably were not sufficiently dedicated in finding these loot boxes, even though they had my screenshots to guide them. This shows the many difficulties with identifying and regulating loot boxes even when experienced playtesters with specialist videogaming knowledge are employed. The IARC was not willing to enter correspondence regarding specific games on the Google Play Store, so there was no opportunity to seek clarity and obtain further rectification.

A closer examination of the rectifications that the IARC did make allowances for interesting reflections on what constitutes a 'loot box' or an 'in-game transaction with randomized elements', as this has been debated academically ([56], cf. [57]) and because these mechanics are difficult to define for regulatory purposes ([40], pp. 351–355). The IARC generally applied the label to so-called social or simulated casino games. This means that, although the paid randomized mechanics contained therein are not necessarily strictly 'loot boxes', they have been deemed to be mechanics involving randomization that should be specifically highlighted to consumers and parents. Since 2020, PEGI gave the PEGI 18 rating to all *new* games containing simulated or actual gambling [27]. It is justifiable for the measure to not be retroactively applied to decades-old physical games, such as *Pokémon Fire Red & Leaf Green Version* (Nintendo, 2004), which contains simulated slot machine gambling [72]. However, PEGI should reconsider whether this measure should be applied retroactively on the Google Play Store to popular social casino games that were released before 2020 but continue to be engaged with by many players today. Applying this policy only to newer games but not older games again constitutes unfair and inconsistent treatment of competitors. The ESRB should consider whether to

R. Soc. Open Sci. **10**: 230270

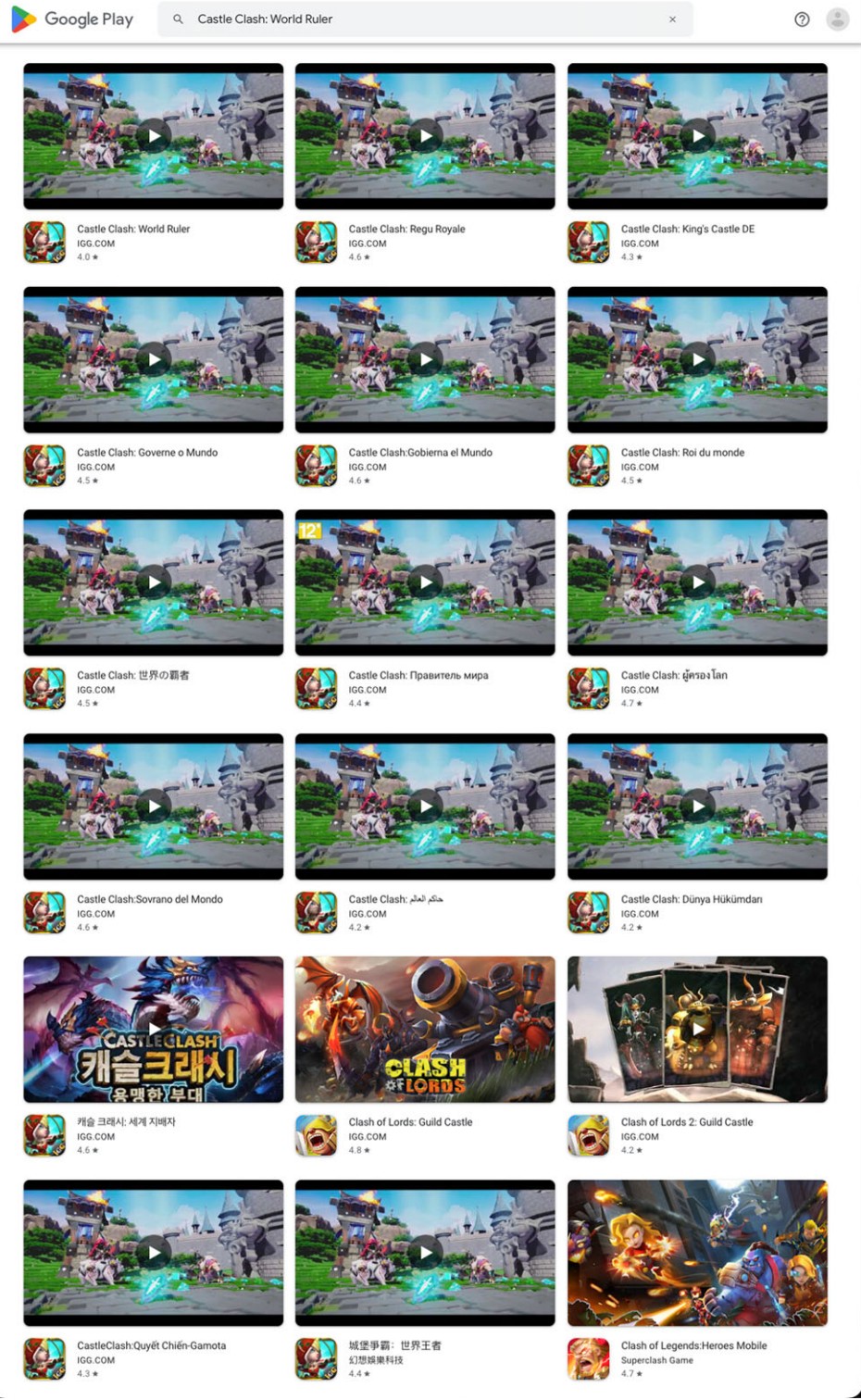

**Figure 4.** Screenshot of the Google Play Store showing search results for 'Castle Clash: World Ruler', which include 15 different release versions of the same game. © 2023 Google; IGG; Superclash Game.

follow PEGI's lead in also imposing a minimum 17+ Mature rating for games involving simulated gambling.

Further, games that contained an energy system (which can be recharged with real-world money) that can be used to complete tasks with random rewards have also been labelled. This shows that although video game companies might be tempted to obfuscate their loot box mechanic (e.g. how the 'virtual

currency' used to 'purchase' the loot box mechanic is represented), a dedicated self-regulator with specialist video game knowledge is willing to and can identify them when called upon to do so.

Although the IARC has since labelled the vast majority of unlabelled games that the present study identified, it has refused to retroactively apply this to other older games that I did not identify. This means that tens of thousands (if not hundreds of thousands) of other games containing loot boxes remain unlabelled on the Google Play Store. Inconsistent retroactive application (i.e. only labelling some historical games but not others) might mislead consumers into falsely believing that all unmarked historical games do not contain loot boxes. As an individual researcher, I could present further lists of games containing loot boxes for labelling by the IARC to chip away at the number of non-compliant games. However, that is not a sustainable, long-term solution, even if more people take up the task. This duty should be the responsibility of the age rating organizations and not volunteers or publicly funded academics. PEGI has stated that the UK Government's technical working group seeking to develop better industry self-regulation of loot boxes has brought together various stakeholders (including Google) and so is optimistic about, and committed to, improving its future approach from a supervisory capacity, while the IARC and the ESRB have made no commitments to improving the future situation and appeared content with the current, objectively unsatisfactory situation.

### 4.2.3. The 'challenging' Google Play Store and Google's failures

PEGI has admitted that the Google Play Store (because of the sheer volume of content available on it) represents a challenge without 'a simple solution' and claimed that other digital storefronts adopting the IARC do not encounter similar difficulties (although this was not independently verified). The ESRB and PEGI (and other age rating organizations) should reconsider whether it is wise to endorse the Google Play Store with the IARC in the current circumstances, given the widespread failure to apply the loot box presence warning label (which may suggest that the self-disclosure of other content has also been left wanting and should be examined by future research). Both organizations have built a credible reputation among consumers, parents and policymakers for historical achievements in relation to video game content moderation. The ESRB and PEGI are risking that goodwill by endorsing the Google Play Store with the IARC. Neither organization can claim to have complete control over the content on the Google Play Store because they only have a supervisory role, and the present study has demonstrated that they do not have broad control even though they might fix individual cases upon request. Both organizations should work with other age rating organizations involved and investigate possibilities of improving the IARC system on the Google Play Store, and if there are no credible paths forward, withdraw their endorsement of the Google Play Store by disapplying the IARC and disallowing the use of the ESRB and PEGI trade dress, so as not to misinform consumers and parents that otherwise trust and rely on the ESRB and PEGI branding.

Indeed, Google, who owns and operates the Google Play Store, should be highlighted as a major perpetrator that has permitted, by omission, this state of non-compliance. Google has not enforced the age rating system (including the label) it supposedly adopted. It is not known whether Google has informed and encouraged companies to add the label since its introduction to the platform. In addition, the IARC label is also not prominently displayed and is difficult to access from either the web or the phone version of the Google Play Store. The label is not automatically shown on the game's page. Using 'Ctrl + F' or otherwise searching for the text of the label would not return any results. The user must always actively do something to cause the label to be displayed. There are two ways to do so on each platform. On the web version, the user must either (a) click on a tiny '(i)' symbol next to the age rating, which causes a pop-up window containing the label to show, or (b) scroll down and click on a right-pointing arrow next to 'About this game', which causes a different pop-up window that the user must then scroll down in order to view the label. On the phone version, the user must perform similar variations of the above-mentioned two methods to cause the label to show. (These are illustrated with screenshots taken of the relevant pages for *Idle Heroes* (DHGAMES, 2016), which are available at the data deposit link.) The Google Play Store has not optimized the visibility of the label to maximize the potential consumer protection benefits of this measure. Google must improve on this. The Italian Competition Authority (Autorità garante della concorrenza e del mercato; AGCM) has enforced the EU Unfair Commercial Practices Directive against Activision Blizzard and Electronic Arts and successfully got both companies to commit to more prominently displaying the PEGI loot box presence warning label on their own websites and storefronts ([13], p. 586). This shows that Google's failure to more conspicuously display the label might be a breach of consumer protection regulation.

Finally, it must also be highlighted that Apple does not even implement a loot box presence warning label on the Apple App Store as part of its own proprietary age rating system [33]. Apple must immediately catch up to the industry standard by adding this feature to better inform consumers. Failure to disclose loot box presence may also contravene consumer protection regulation.

### 4.2.4. *Roblox* and *Minecraft*

Besides the sample of 100 random games, *Roblox* and *Minecraft* were separately analysed due to them being unique games where third-party user-generated content (including loot boxes) can be created and published, and then subsequently bought and sold in exchange for real-world money. Neither game displayed the loot box presence warning label on the Google Play Store as of 12 January 2023.

On 30 September 2022, PEGI recognized that 'Roblox is considered as a platform with diverse content rather than an individual game product' and announced that it had changed the rating for *Roblox* from 'PEGI 7' to the 'Parental Guidance' label (which is usually reserved for non-video gaming apps that act like platforms for 'a broad, variable selection of content', such as Netflix and Spotify) due to 'the large variety of user-generated content that is available in Roblox' [73]. This occurred on the same date that the BBC reported on *Roblox*'s removal of user-generated content depicting contemporaneously ongoing real-world armed conflicts and commercially profiting from depictions thereof [74]. The PEGI announcement also referred to *Roblox*'s parental tools; however, *Roblox*'s internal 'age recommendations and content descriptors' system does not make reference to loot boxes (or any similar paid mechanic involving randomization) or take them into account when determining the appropriate 'age recommendations' for user-generated content [75]. *Roblox* provides other parental tools, such as setting monthly spending restrictions (but only generally, as a limit cannot be set on loot box spending specifically) [76]; however, no parental resources discuss loot boxes or can be used to address loot box-related concerns specifically [77–79].

The ESRB has not officially rated *Roblox* itself (as *Roblox* is a digitally released game without a corresponding physical release and so the IARC system suffices), but reportedly and confirmed through the present analysis, the ESRB age rating obtained and shown through IARC also changed from 'Everyone 10+' to 'Teen for Diverse Content: Discretion Advised' (i.e. suitable for 13+ [28]) at the same time as the PEGI age rating change [80].

*Roblox* knows that user-generated content within its ecosystem contains loot boxes, as demonstrated, for example, by how it has required loot box probability disclosures in relation to such third-party content since 2019 [55] (probably to ensure, or at least project, its own compliance with the Apple App Store's and the Google Play Store's platform-wide, self-regulatory requirement that probability disclosures must be made [18]). In addition, *Roblox* has reportedly led a programme of complying with Dutch and Belgian gambling laws (the latter of which forbids all paid loot boxes [35], while the exact legal position of the former is less clear following a court decision in March 2022 [81]) by removing loot boxes from those two jurisdictions, following the publication of the author's previous study on Belgium's largely unenforced and ineffective ban on loot boxes [49]. Therefore, *Roblox* is fully aware of the presence of loot boxes within its game through third-party user-generated content and should know that the loot box presence warning label must be attached. It can be fairly assumed that Roblox Corporation did not answer the relevant question on the IARC questionnaire accurately.

By contrast to *Roblox*, *Minecraft* has little published information on the loot boxes that are available through third-party user-generated content available on its platform. However, similarly to Roblox Corporation, the operator of *Minecraft*, Mojang Studios (which is a subsidiary of Xbox Game Studios, which in turn is a subsidiary of Microsoft) should be treated as having constructive, if not actual, knowledge of the presence of loot boxes within the game. Mojang Studios was similarly obliged to answer the relevant IARC question correctly and thereby ensure that the label was appended to *Minecraft*.

*Roblox* and *Minecraft* should now immediately amend their respective answers to the IARC questionnaire and attach the loot box presence warning label. As loot boxes are of particular concern to consumers, including parents, as the ESRB has previously publicly recognized when introducing the label [3] (although the ESRB has since denied that parents were concerned about loot boxes in its communications with me). Therefore, in addition, both games should make available parental resources that address loot boxes specifically, rather than just in-game spending generally: this should be in the forms of both (i) advisory information to highlight the loot box issue and (ii) control tools that can be used to actively intervene if the parent deems doing so necessary. To facilitate and enforce internal platform-wide self-regulation, third-party providers of user-generated content should be required by *Roblox* and *Minecraft* to prominently disclose the presence of loot boxes on the relevant platform store pages.

Lastly, *Roblox* and *Minecraft* can consider the 'nuclear' option of banning loot boxes from the games/platforms entirely. Rockstar Games recently announced in November 2022 that, in relation to 'third party "Roleplay" servers' for its popular *Grand Theft Auto* (1997–present) and *Red Dead Redemption* (2010–present) series, it forbids and will take 'legal enforcement' action against 'commercial exploitation, including the sale of 'loot boxes' for real-world currency or its in-game equivalent' [82].[2] Government authorities in Belgium may have failed to eradicate loot boxes on a national level due to a lack of resources and enforcement [35]. However, *Roblox* and *Minecraft* are platforms that can more directly and effectively regulate and enforce rules. These two games are in a position to lead the industry in terms of loot box regulation (and, if deemed appropriate, elimination). If successful, self-regulation by these two games can become a proof of concept for other bigger platforms, such as the entire Apple App Store and Google Play Store. Countries can also consider establishing legal regulation (thus not leaving the rule-making in the hands of arguably conflicted private organizations, as is the case with industry self-regulation) but placing the burden of enforcing those legal rules on the platforms, who can probably assess non-compliance and impose sanctions better than national gambling regulators [35].

The IARC refused to label *Minecraft* and *Roblox* when I asked. In relation to *Minecraft*, the IARC denied that this game contains loot boxes. This is a debatable point. One comment written by a player in response to a news report of the present study also expressed doubt as to whether *Minecraft* 'contains loot boxes' *per se*; however, this comment also pointed out that some third-party servers of *Minecraft* do sell loot boxes through their own separate stores [83]. If Rockstar Games knows about loot boxes in the third-party servers of *Grand Theft Auto* and *Red Dead Redemption*, then Mojang Studios and Microsoft should know about the same occurring in *Minecraft*'s third-party servers. The Rockstar games are generally rated suitable only for adults, while *Minecraft* is rated suitable for young people aged 7 or 10+, so Mojang Studios and Microsoft should be acting even more proactively than Rockstar Games at protecting their younger and more vulnerable players. In relation to *Roblox*, the IARC claimed that '*Roblox* is not rated as a game and as such would not qualify for [the label]'. With respect, *Roblox* is made available on the Google Play Store under the 'Games' category. The Google Play Store page for *Roblox* has a heading entitled 'About this *game*' (emphasis added). If it looks like a game, plays like a game and is advertised as a game, then it probably is a game. Young people and parents would treat *Roblox* as a video game: the IARC is ducking its responsibilities by permitting *Roblox* to masquerade as 'not a game' and escape liability. Instead, *Roblox* should be self-regulated both as a game *and* as a 'platform with diverse content' (as PEGI puts it [73]) to properly reflect the compounded potential dangers it presents.

## 4.3. Developing better loot box warnings

It is important for a warning to provide sufficient information without unfairly exaggerating the harms of the underlying product, unless the latter is explicitly intended by policymakers to discourage engagement with the product (such that what might have been a non-intrusive 'nudge' in the behavioural intervention context becomes a more coercive 'shove').

### 4.3.1. Insufficient information and unclarity

Criticisms of the ESRB, PEGI and IARC label of 'In-Game Purchases (Includes Random Items)' have already been made both theoretically after a plain reading of the text [23] and empirically under experimental conditions [31]. It does not provide enough information to help consumers to identify exactly where the loot boxes can be found in the game, such that consumers might be enabled to recognize them and choose to not engage with them or to prevent their children from engaging with them ([23], pp. 2358–2359). For all other potentially problematic content, PEGI has a graphic label that attempts to convey the relevant issue (e.g. a clenched fist to show 'Violence'); however, the loot box presence warning label only appears in plain text and is less visually prominent than other content descriptors, thus arguably suggesting that it is less concerning than other issues (figure 2). Further, under experimental conditions, 'consumers do not appear to understand the ESRB/PEGI loot box warning', ([31], p. 1) possibly due to the label's use of newly coined terminology: 'Random Items' or 'Paid Random Items' (which PEGI has since abandoned) were artificially invented as they were not used colloquially to refer to loot boxes and similar mechanics prior to the labels being introduced,

---

[2]Rockstar Games' use of 'commercial exploitation' is probably only in the sense of generating revenue, rather than a value judgement on the nature of the loot box monetization model itself.

and these terms do not appear to have been adopted by player communities and the wider public discourse. Finally, these warnings do not need to be attached (and therefore can be circumvented) if the game offers no digital loot boxes but instead offers physical packs of random cards for sale that do have digital functionality within the game that allows the physical card packs to effectively act like loot boxes, as was the case with packs of randomized amiibo cards intended for use in *Animal Crossing: New Horizons* (Nintendo, 2020) ([23], p. 2359; [84], pp. 31–32).

Henceforth, it appears reasonable to suggest that the warning label should be updated to explicitly refer to both 'loot boxes' and 'gacha', which are two of the most popular terms presently used colloquially to describe 'in-game transactions with randomized elements'. Empirical studies should be conducted to determine what terms are most widely understood by various stakeholders, particularly young players and parents: '[Paid] Random Items' is likely to be one of the least widely understood. However, references to 'loot boxes' and 'gacha' should be made in a non-exhaustive way so as not to incorrectly imply that these are the only two forms of such mechanics [40]. An improved warning should perhaps read 'In-Game Purchases (Includes Loot Boxes, Gacha, or Other Products Offering RANDOM Results)'.

### 4.3.2. Avoid fear-mongering

On the other hand, the warning label that was under consideration at the Illinois state legislature quoted in full in the Introduction section implies causation in the sense that loot box purchasing would lead to gambling addiction. That warning also reads very similarly to pre-existing alcohol and tobacco warnings ([31], p. 12), which might cause readers to draw parallels between these products and which might then, by making this allusion, cause readers to have an overly negative perception of loot boxes. These issues meant that the Illinois warning might *overstate* the potential harms of loot boxes ([28], p. 12), in contrast to the ESRB/PEGI/IARC label which *understates* them ([31], p. 10).

To create the most appropriate warning, a balance must be struck between providing enough relevant information and not overemphasizing potential harms. Any warning should be subjected to testing prior to being adopted. Neither the ESRB nor PEGI has claimed that their labels have been tested and certified as being effective at informing consumers about loot boxes' potential harms prior to their introduction, and independent testing by academics suggests that they fail to do so [31].

### 4.3.3. Lack of uniformity

When the labels were first introduced on 13 April 2020, the ESRB label read 'In-Game Purchases (Includes Random Items)', while the PEGI label had 'IN-GAME PURCHASES' as a graphic symbol in addition to the text '(Includes Paid Random Items)'. Textually, these were largely identical but for PEGI's addition of 'Paid'. However, technically, they were not exactly the same. The rationale behind this difference has not been publicly explained (which should still be done for transparency). The PEGI label might have been superior as it re-emphasized the 'paid' nature of these in-game purchases involving random items (thus differentiating them from other non-paid video game mechanics involving randomization), in which case the ESRB label should have been changed, although stating 'paid' after 'in-game purchases' appears repetitive. This lack of uniformity would have been problematic, but this difference has since been resolved (perhaps in the wrong direction) because the PEGI label was changed without announcement to be the same as the ESRB label on an undeterminable date lying between 13 June 2020 (https://web.archive.org/web/20200613021408/https://pegi.info/news/pegi-introduces-feature-notice) and 12 July 2020 (https://web.archive.org/web/20200712034753/https://pegi.info/news/pegi-introduces-feature-notice), as respective web page snapshots made by the archival Wayback Machine demonstrate. The PEGI original announcement of the label has been partially amended: as of 16 January 2023, the text still refers to the outdated '(Includes Paid Random Items)' wording but the accompanying images show the 'In-game Purchase (Includes Random Items)' wording instead [24].[3]

In the IARC context, which label is used continues to be relevant because Germany's USK age rating organization has recently introduced its own new loot box presence warning label of 'In-Game-Käufe + zufällige Objekte [In-game purchases + random items]' [38], which is quite different from the 'In-Game-

[3]The ESRB's capitalization of the 'G' in 'In-Game', which PEGI has not done, is assumed to be trivial. Indeed, on the inconsistent capitalization point, in the PEGI search tool, the label is shown as 'In-game Purchases (includes random items)' with the latter three words uncapitalized, contrary to what is written elsewhere (e.g. on physical game boxes).

Käufe (zufällige Objekte möglich) [In-Game Purchases (Random Items Possible)]' presently used by PEGI in the German language [85]. The IARC has been updated in the German context, as of 17 January 2023 (if not earlier), to use the USK label rather than the ESRB/PEGI label, as demonstrated by a printout of the updated German Google Play Store page for *Guns of Glory* (available at the data deposit link). Perhaps both labels should instead have been retained for clarity. As an aside, the policy recommendations in the preceding paragraphs regarding the ESRB/PEGI label are also applicable to the USK's new label. Closer examination of the German PEGI label (which the IARC now no longer uses) also reveals that it would be translated into English as 'In-Game Purchases (Random Items *Possible*)' (emphasis added) rather than 'In-Game Purchases (Includes Paid Random Items)', which is the original English version. A slightly different meaning is expressed by the German label as compared with the original English label. This reflects that in addition to testing the English version of the label for efficacy, other language versions of the label would also need to be separately tested (i.e. the Spanish 'Compras dentro del juego (Incluye artículos aleatorios)' [86] used by the ESRB and the 24 non-English language versions used by PEGI).

### 4.3.4. Toward an internationally uniform label

Uniformity of the loot box presence warning label internationally has merit. Consumers would be able to recognize this information even when travelling outside of their home jurisdiction, and companies would be able to comply more easily and efficiently as only one label (or at least only different language versions of one label) needs to be incorporated into product design. Streamlining, providing consistent information to consumers, and making it easier to conduct business internationally were some of the founding principles of the IARC system [32]. If that system could be agreed, then why could there not be a uniform loot box presence warning label? One counter-argument might be that dedicated language labels might be more informative to certain consumers, but that point remains to be empirically proven and, even after it has been proven, that one advantage must be balanced against the many advantages of an internationally uniform label. PEGI changing its label to match the ESRB's is a step in the right direction (at least in terms of uniformity, if not efficacy), while the USK's introduction of a new and somewhat different label (whose efficacy has probably also not been tested) should be criticized. Rather than introducing even more types of labels, regulators in various jurisdictions (both legislatures and self-regulatory age rating organizations) should work together to develop an effective and uniform loot box presence warning label. As part of that new label, regulators should consider disclosing more information, such as providing a link to the loot box probability disclosures, which are now generally required either by law or industry self-regulation across the world but have proven difficult for consumers to access [18,20]. Consumers should also be told exactly where loot boxes can be found in-game; how much they cost; whether they provide competitive advantages; how to turn off the ability to purchase them, etc. The final design of the label should also be kept updated to address new concerns as they arise.

### 4.4. Limitations

For Study 1, reliance was placed on the ESRB's and PEGI's official age rating search tools. Relying on these accredited online resources that are intended for players and parents to place their trust in and use is fair in the circumstances. However, it is nonetheless possible that data entries on these official tools may be incorrect, incomplete and inaccurate. For example, when comparing the results from two separate occasions when data were scraped (including for an exploratory analysis that is no longer part of this paper), one ESRB game was seemingly removed from the search tool. In particular, it is unlikely but possible that certain entries were missing from both the ESRB and PEGI Lists and therefore were not included within the ambit of the present study: this could be (i) due to a certain game not reporting loot box presence to both age rating organizations and this not being detected, such that the game was never correctly labelled by either, or (ii) even though a certain game did accurately report loot box presence and it was labelled as such, both systems had data entry errors and failed to correctly note this on their search tools. Further, the accuracy of the data scraping for Study 1 was manually verified for a limited number of entries; however, it remains possible that the data scraping was not perfectly accurate. It should also be noted that, even if the labels were accurately attached to the relevant games by the ESRB and PEGI and were so shown on the online search tool, it remains possible that during the production process the labels were not accurately attached to physical products and digital storefronts.

Study 1 could not consider the practical application of these labels, specifically, whether their use in real-life has been accurate. To illustrate, in relation to *Genshin Impact*, the author has received marketing emails from the operating company, HoYoverse (which is a more recent rebranding of the company that miHoYo uses outside of China [87,88]), advertising new in-game content. One such email received on 13 January 2023 has been archived at the data deposit link. Both the ESRB and PEGI age ratings were appended at the bottom of said email. However, the loot box presence warning label was attached to neither. This is despite PEGI having labelled the game (the ESRB has not at the time). This non-compliant and inaccurate marketing email appended the PEGI age rating *without* the warning label despite PEGI having attached it. PEGI has explained that this is not in breach of the *PEGI Code of Conduct* because accurate labelling obligations did not extend to promotional materials [64]. However, PEGI has promised that its guidelines will be updated to ask publishers to include the label for promotional materials (although this might not include targeted marketing emails sent to already registered users). This case demonstrates how in practice the labels might not be shown to consumers by game companies even when it has been correctly applied for from, and given by, the relevant age rating organization. This issue should be the subject of future research.

Finally, the German age rating organization, the USK, announced that it will attach its loot box presence warning label of 'In-Game-Käufe + zufällige Objekte [In-game purchases + random items]' after the present study was planned and proposed [38]. Given the dates at which the present study was planned to be conducted, not enough time would have passed for the USK to have labelled many games (if any at all), and the USK stated that it would not retroactively apply the label, so it was deemed inappropriate to include the USK in the present study. A replication of Study 1 after some time has passed should include the USK as an additional comparator with the ESRB and PEGI.

For Study 2, only the compliance situation on the Google Play Store for mobile Android games was assessed: other participating storefronts for PC and console game platforms, such as the Microsoft Store for Windows and Xbox, the Nintendo eShop and the PlayStation Store [32], may exhibit different compliance behaviours, similarly to how the prevalence rate of loot boxes differs significantly between the PC and mobile platforms ([34], p. 1770). PEGI has expressed the view that the compliance situation on those other platforms would be significantly better. However, this assertion was not independently verified and should be the subject of future research. In addition, the sample's representativeness is constrained by practical reasons, similarly to previous loot box prevalence studies. As explained in the Method section, the sample was formed of previously popular and high-grossing games. The compliance situation among this sample is not necessarily representative of the whole Google Play Store (e.g. how a sample of randomly selected Google Play Store games regardless of financial performance would have complied). The present sample was probably more compliant than average because more popular and higher-grossing games are probably operated by companies that have more resources and are more heavily scrutinized and frequently monitored for compliance by players, parents, competing companies, regulators and other stakeholders. Conversely, it is also possible (albeit highly unlikely) that the loot box presence warning label is effective at reducing spending (which empirical evidence does not support [31]), such that more-compliant games performed worse financially, and the higher-grossing games made more money because of their non-compliance. Regardless, the present sample's results are still informative and relevant because stakeholders (players, parents and policymakers) would probably be more interested in whether these popular games were complying rather than whether unknown, poorly performing games were complying.

Finally, the author's own interpretation of the official responses by PEGI and the ESRB/the IARC was presented. Readers are invited to form their own opinions by perusing those responses, which have been made publicly available at the data deposit link.

# 5. Conclusion

The present study assessed compliance with the ESRB's, PEGI's and IARC's loot box presence warning label of 'In-Game Purchases (Includes Random Items)' through two studies. Study 1 found that, as to physical games rated by the ESRB and PEGI, there were many instances (60.6% of all games labelled by either age rating organization or 16.1% using a more equitable methodology) where the two organizations have been inconsistent and not both applied the label to the same game. The vast majority of those inconsistencies were caused by the ESRB not retroactively labelling older games, which PEGI has done with ease and at minimal costs. The ESRB has refused to emulate PEGI's better

approach. Four cases where the ESRB and one case where PEGI culpably failed to label a game were identified. The ESRB admitted fault in relation to one game and refused to admit fault in relation to three other games, even though those failings arose from the same circumstances as the one case for which PEGI admitted fault and committed to improving. Overall, PEGI's implementation of the label is reasonably satisfactory given its proactive retroactive application and demonstrable willingness to do even better. By contrast, the ESRB's implementation is less satisfactory: because many older games are not, and will not, be labelled, the measure could not be relied upon by consumers and parents to provide accurate information in relation to historical games that remain popular presently. The ESRB must also be criticized for being unreceptive to practicable suggestions that would improve its procedure. However, in relation to newly released physical games for console/PC platforms, the labelling at both PEGI and the ESRB should be reasonably accurate and reliable.

Study 2 found that, as to digital games rated through the IARC, most games (71.0%) containing loot boxes on the Google Play Store did not accurately display the label. Most of the identified non-compliant games have since been labelled through rectifications at the author's request. However, the IARC generally denied liability (unconvincingly) by stating that older games submitted for rating prior to February 2022 are not required to display the label and has refused to apply the label to older games beyond the sample that the author has identified (which represents only a tiny proportion of all games containing loot boxes on the Google Play Store). At present, this self-regulatory measure cannot be treated as a trustworthy and dependable authentication of whether a game contains loot boxes on the Google Play Store. PEGI has admitted that the Google Play Store poses a 'challenge' that presently does not have 'a simple solution'. Consumers, parents, regulators and all other stakeholders should rely on the label cautiously: a game marked with the label will contain loot boxes; however, a game not thusly labelled may also contain loot boxes. At present, this measure fails to provide accurate information to consumers. The mere existence of this measure cannot be used to justify the non-regulation of loot boxes, given the poor compliance and doubtful efficacy (even if the measure is complied with satisfactorily). In addition, this measure (or an equivalent) is not implemented on the Apple App Store. Currently, consumers are not being provided with adequate information about loot box presence on the two major mobile app stores and the mobile platform generally.

Age rating organizations are expected to, either directly (i.e. the ESRB and PEGI in Study 1 and the IARC in Study 2) or indirectly (i.e. the ESRB and PEGI in Study 2), provide effective and accurate information and content moderation. The ESRB and PEGI promised to label games containing loot boxes on the Google Play Store by endorsing the IARC, but the IARC has demonstrably failed to do so. This ill-advised endorsement caused them to betray the trust placed in them by consumers, parents, and policymakers that rely on them to make informed purchasing decisions and self-regulate the industry. Most high-grossing games were released prior to February 2022, and this is unlikely to change for years to come. Stakeholders would naturally expect, and should demand, the measure to be applied retroactively. Otherwise, this measure would have little practical benefit (besides falsely demonstrating the industry supposedly taking action to address loot box harms). The existing system must be improved upon: loot box warning labels should be applied retroactively, as the minimal additional compliance costs are justified. Age rating organizations should collaborate and cross-check each other's labelling to correct mistakes and enhance accuracy. The IARC rating system relies solely on self-disclosure, which has demonstrably been inadequate. In relation to the highest-grossing games, the IARC ought to involve additional external scrutiny that seeks to verify the self-disclosures' accuracy and completeness. The design and consumer protection efficacy of the label in practice (e.g. whether it is well understood by parents) probably also requires improvement. Regulators should strive toward developing a uniform and effective label that provides sufficient information without overexaggerating potential harms.

**Intellectual property notice.** As of 12 January 2023, the access, extraction, publication and use of the ESRB and PEGI age rating archives for, *inter alia*, academic research and criticism purposes in the public interest are reasonably assumed to be lawful, particularly considering that publication has been limited only to relevant materials. If copyright and/or database rights subsist in the ESRB and PEGI rating archives, then, having acknowledged the relevant sources, the author uses such data under relevant fair use/fair dealing and 'permitted acts' provisions of copyright law and database rights regulations, as applicable.

**Positionality statement.** In terms of the author's personal engagement with loot boxes, he plays video games containing loot boxes, but he has never purchased any loot boxes with real-world money.

**Article history.** Stage 1 Registered Report Recommendation: https://rr.peercommunityin.org/articles/rec?id=317.
  Stage 2 Registered Report Recommendation: https://rr.peercommunityin.org/articles/rec?id=404.

This Registered Report was submitted to Royal Society Open Science following peer review and recommendation for Stage 2 acceptance at the Peer Community In (PCI) Registered Reports platform. Full details of the peer review and recommendation of the paper at PCI Registered Reports may be found at the links above.

After submission to the journal, the paper received no additional external peer review, but was accepted on the basis of the Editor's recommendation according to our PCI Registered Reports policy https://royalsociety publishing.org/rsos/registered-reports#PCIRR.

**Data accessibility.** The underlying data, a full library of PDF printouts and screenshots showing, *inter alia*, the relevant Google Play Store web page sections and in-game loot box purchase pages for each game, and the official responses from PEGI and the ESRB/the IARC are publicly available in the Open Science Framework at https://doi.org/10.17605/OSF.IO/YZKUP [89].

The PCI RR Study Design Table for the present study is provided as electronic supplementary material [90].

**Author's contributions.** L.Y.X.: conceptualization, data curation, formal analysis, investigation, methodology, project administration, resources, software, visualization, writing—original draft, writing—review and editing.

**Conflict of interest declaration.** After the first preprint version of the study was published on 18 January 2023, L.Y.X. has communicated in writing with PEGI and the ESRB/the IARC by email and has met with PEGI in a remote meeting on 20 January 2023. All written communications are available at the data deposit link. L.Y.X. was employed by LiveMe, then a subsidiary of Cheetah Mobile (NYSE:CMCM), as an in-house counsel intern from July to August 2019 in Beijing, People's Republic of China. L.Y.X. was not involved with the monetization of video games by Cheetah Mobile or its subsidiaries. L.Y.X. undertook a brief period of voluntary work experience at Wiggin LLP (Solicitors Regulation Authority (SRA) number: 420659) in London, England in August 2022. L.Y.X. has met and discussed policy, regulation and enforcement with the Belgian Gaming Commission [Belgische Kansspelcommissie] (June 2022 & February 2023), the Danish Competition and Consumer Authority [Konkurrence- og Forbrugerstyrelsen] (August 2022) and the Department for Digital, Culture, Media and Sport (DCMS) of the UK Government (August 2022). L.Y.X. has been invited to provide advice to the DCMS on the technical working group for loot boxes and the Video Games Research Framework. L.Y.X. was the recipient of two AFSG (Academic Forum for the Study of Gambling) Postgraduate Research Support Grants that were derived from 'regulatory settlements applied for socially responsible purposes' received by the UK Gambling Commission and administered by Gambling Research Exchange Ontario (GREO) (March 2022 and January 2023). L.Y.X. has accepted funding to publish academic papers open access from GREO that was received by the UK Gambling Commission as above (October, November and December 2022). L.Y.X. has accepted conference travel and attendance grants from the Socio-Legal Studies Association (February 2022 and February 2023), the Current Advances in Gambling Research Conference Organizing Committee with support from GREO (February 2022), the International Relations Office of The Jagiellonian University (Uniwersytet Jagielloński), the Polish National Agency for Academic Exchange (NAWA; Narodowa Agencja Wymiany Akademickiej) and the Republic of Poland (Rzeczpospolita Polska) with co-financing from the European Social Fund of the European Commission of the European Union under the Knowledge Education Development Operational Programme (May 2022), and the Society for the Study of Addiction (November 2022). L.Y.X. was supported by academic scholarships awarded by The Honourable Society of Lincoln's Inn (March 2020) and The City Law School, City, University of London (July 2020).

**Funding.** L.Y.X. is supported by a PhD Fellowship funded by the IT University of Copenhagen (IT-Universitetet i København), which is publicly funded by the Kingdom of Denmark (Kongeriget Danmark).

**Acknowledgement.** Thanks to David Zendle for inspiring this study, discussing potential methodologies with the author, and graciously allowing the author to pursue this project independently. Credit is also due to all the co-authors of Zendle *et al.* (2020) for making the underlying data publicly available for further study and reanalysis [34]. Thanks to Rune Kristian Lundedal Nielsen, Laura L. Henderson and Pieterjan Declerck for helpful comments on earlier drafts of this manuscript. Thanks to Aaron Drummond, Pete Etchells and Chris Chambers for valuable feedback during the review process. Thanks to the staff at Manchester Metropolitan University and SCONUL (the Society of College, National and University Libraries) for facilitating my library access, which ensured speedy data collection. Thanks to Christopher Lukman for assistance with the German language issue.

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
