## [Peer Review File · Royal Society Open Science]

Review History

Decision letter (RSOS-230270.R0)

Dear Mr Xiao

On behalf of the Editor, I am pleased to inform you that your Manuscript RSOS-230270 entitled "Beneath the label: Unsatisfactory compliance with ESRB, PEGI, and IARC industry self-regulation requiring loot box presence warning labels by video game companies" has been accepted in principle for publication in Royal Society Open Science.

You may now progress to Stage 2 and complete the study as approved. **Please note, there is no action required from you at this stage, and we will create the Stage 2 RR submission in the system on your behalf.**

If you have any questions at all, please do not hesitate to get in touch.

on behalf of Professor Chris Chambers (Associate Editor) and Chris Chambers (Registered Reports Editor, Royal Society Open Science)
openscience@royalsociety.org

Author's Response to Decision Letter for (RSOS-230270.R0)

See Appendix A.

Decision letter (RSOS-230270.R1)

Dear Mr Xiao:

I am pleased to inform you that your Stage 2 Registered Report entitled "Beneath the label: Unsatisfactory compliance with ESRB, PEGI, and IARC industry self-regulation requiring loot box presence warning labels by video game companies" is now accepted for publication in Royal Society Open Science via the PCI RR submission track.

Please remember to make any data sets or code libraries 'live' prior to publication, and update any links as needed when you receive a proof to check - for instance, from a private 'for review' URL to a publicly accessible 'for publication' URL. It is also good practice to add data sets, code and other digital materials to your reference list.

Royal Society Open Science is a fully open access journal. A payment may be due before your article is published. Please note that, if the corresponding author of your paper is based at an institution covered by one of our Transformative Agreement deals, your fees may be covered by the deal - please check the list of eligible institutions at <https://royalsociety.org/journals/authors/read-and-publish/read-publish-agreements/>. The Royal Society has partnered with Copyright Clearance Center's (CCC's) RightsLink service to allow authors to pay article processing charges or page charges. After your manuscript has been accepted, the corresponding author will receive an email from CCC with the subject "Please submit your article processing/open access charge(s)/page charges" inviting you to pay your charges or request an invoice. The email from CCC will come from the email domain @copyright.com (if you have any queries regarding fees, please see <https://royalsocietypublishing.org/rsos/charges> or contact authorfees@royalsociety.org). If you request an invoice, it will be sent to you from CCC. It is important to be cautious about payment scams. **If you receive an email or text message requesting payment and have any concerns, we recommend contacting us through our website, rather than clicking on any links. The Royal Society will never ask you to make a direct payment.**

The proof of your paper will be available for review using the Royal Society online proofing system and you will receive details of how to access this in the near future from our production office (openscience_proofs@royalsociety.org). We aim to maintain rapid times to publication after acceptance of your manuscript and we would ask you to please contact both the production office

and editorial office if you are likely to be away from e-mail contact to minimise delays to publication. If you are going to be away, please nominate a co-author (if available) to manage the proofing process, and ensure they are copied into your email to the journal.

on behalf of Professor Chris Chambers (Subject Editor).

Follow Royal Society Publishing on Twitter: @RSocPublishing
Follow Royal Society Publishing on Facebook:
<https://www.facebook.com/RoyalSocietyPublishing/>
Read Royal Society Publishing's blog:
<https://royalsociety.org/blog/blogsearchpage/?category=Publishing>

Appendix A

Stage 1 Registered Report Recommendation:

<https://rr.peercommunityin.org/articles/rec?id=317>.

Stage 2 Registered Report Recommendation:

<https://rr.peercommunityin.org/articles/rec?id=404>.

This Registered Report was submitted to Royal Society Open Science following peer review and recommendation for Stage 2 acceptance at the Peer Community In (PCI) Registered Reports platform. Full details of the peer review and recommendation of the paper at PCI Registered Reports may be found at the links below.

After submission to the journal, the paper received no additional external peer review, but was accepted on the basis of the Editor's recommendation according to our PCI Registered Reports policy

<https://royalsocietypublishing.org/rsos/registered-reports#PCIRR>.